# Phase separation of protein mixtures is driven by the interplay of homotypic and heterotypic interactions

Mina Farag [1], Wade M. Borcherds [2,3], Anne Bremer[2,3], Tanja Mittag [2] ✉ & Rohit V. Pappu [1] ✉

Prion-like low-complexity domains (PLCDs) are involved in the formation and regulation of distinct biomolecular condensates that form via phase separation coupled to percolation. Intracellular condensates often encompass numerous distinct proteins with PLCDs. Here, we combine simulations and experiments to study mixtures of PLCDs from two RNA-binding proteins, hnRNPA1 and FUS. Using simulations and experiments, we find that 1:1 mixtures of A1-LCD and FUS-LCD undergo phase separation more readily than either of the PLCDs on their own due to complementary electrostatic interactions. Tie line analysis reveals that stoichiometric ratios of different components and their sequence-encoded interactions contribute jointly to the driving forces for condensate formation. Simulations also show that the spatial organization of PLCDs within condensates is governed by relative strengths of homotypic versus heterotypic interactions. We uncover rules for how interaction strengths and sequence lengths modulate conformational preferences of molecules at interfaces of condensates formed by mixtures of proteins.

Biomolecular condensates are membraneless bodies that provide spatial and temporal control over cell signaling and cellular responses to various stresses[1–7]. Functional roles for condensates have been implicated in transcriptional regulation[8–14], cytosolic and nuclear stress responses[3,15–21], trafficking of cellular components[22,23], RNA regulation and processing[24–28], mechanotransduction[29–31], and protein quality control[32–36]. The working hypothesis, based on a growing corpus of data, is that condensates form via spontaneous and driven phase transitions of networks of multivalent biomacromolecules[1,37,38]. The relevant processes involve a coupling of associative and segregative phase transitions[39]. These include processes such as phase separation coupled to percolation (PSCP)[39–41] and complex coacervation[42–44]. Multivalent proteins that scaffold and drive phase transitions encompass different numbers and types of oligomerization and substrate binding domains[41]. Most, although not all protein scaffolds also feature intrinsically disordered regions (IDRs) that drive or modulate phase separation of

protein scaffolds through a blend of homotypic and heterotypic interactions[45]. Here, homotypic, and heterotypic interactions refer to intermolecular interactions between the same versus different molecules, respectively. This concept can be extended to distinguish interactions between the same versus different motifs on molecules[46].

IDRs that drive condensate formation include a family of sequences referred to as prion-like low-complexity domains (PLCDs)[47,48]. These domains are readily recognizable based on their compositional biases[49]. Our analysis of 89 condensate-associated PLCDs drawn from the human proteome shows that on average, 50–60% of the amino acid residues within PLCDs are polar (Gly, Ser, Thr, Gln, and Asn), ~10% are aromatic (Tyr, Phe, Trp, and His), ~13% are Pro, fewer than 5% are charged, and the remaining residues are aliphatic (Fig. 1a).

Recent studies have focused on quantitative assessments of how distinct sequence features contribute to the driving forces for phase

[1]Department of Biomedical Engineering and Center for Biomolecular Condensates, Washington University in St. Louis, St. Louis, MO 63130, USA. [2]Department of Structural Biology, St. Jude Children's Research Hospital, Memphis, TN 38105, USA. [3]These authors contributed equally: Wade M. Borcherds, Anne Bremer. ✉e-mail: tanja.mittag@stjude.org; pappu@wustl.edu

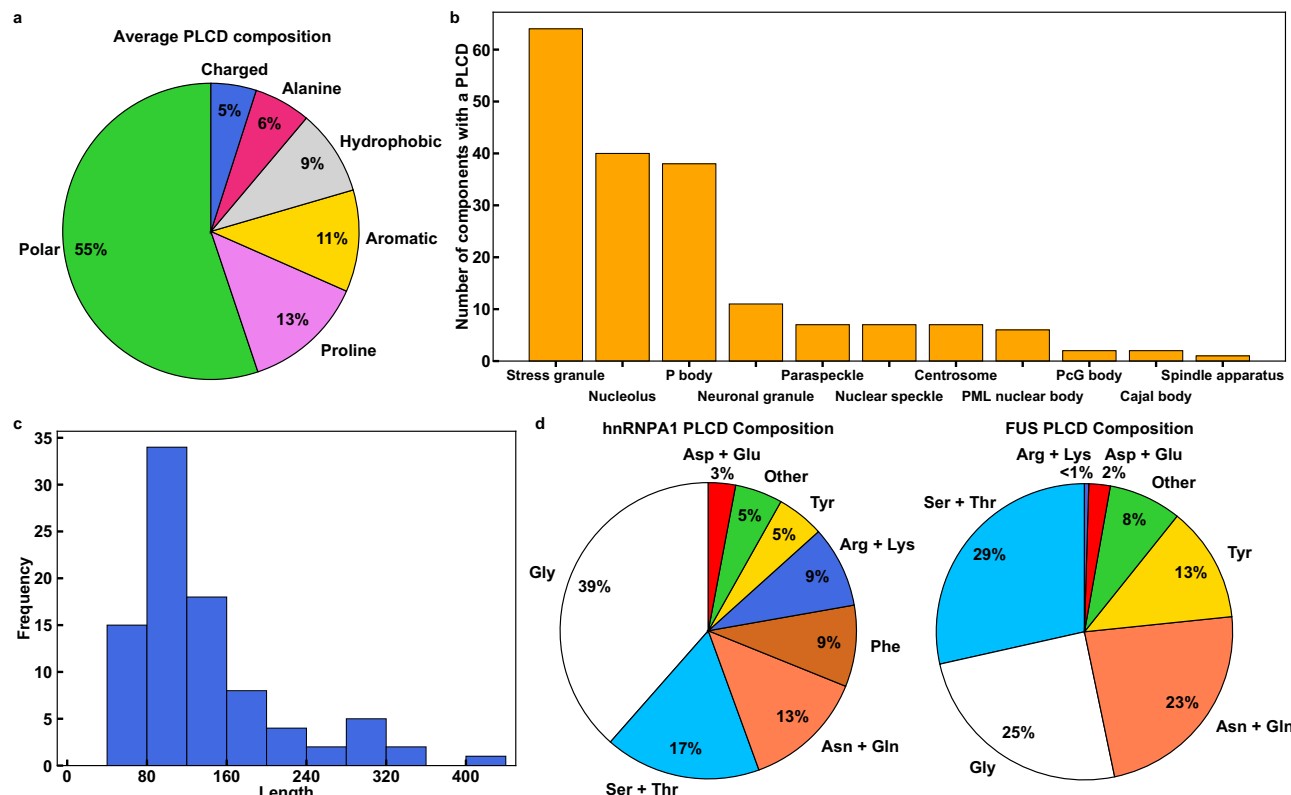

**Fig. 1 | PLCDs of similar albeit non-identical compositions are prevalent in distinct biomolecular condensates. a** Average compositional profile of 89 PLCDs found within cellular condensates in *Homo sapiens*. Here, "Polar" includes Gly, Ser, Thr, Asn, Gln, and Cys. "Aromatic" includes Phe, Tyr, Trp, and His. "Hydrophobic" includes Leu, Iso, Val, and Met. "Charged" includes Asp, Glu, Arg, and Lys. **b** Distribution of PLCDs across cellular condensates in *Homo sapiens*. The condensates examined here have at least 40 known protein components. **c** Length distribution of all 89 distinct PLCDs in **a** and **b**. **d** Compositional profiles of PLCDs within human versions of hnRNPA1 and FUS. For the hnRNPA1 PLCD, the "Other" category includes 2.2% Ala, 1.5% Met, and 1.5% Pro. For the FUS PLCD, the "Other" category includes 1.9% Ala, 1.0% Met, and 5.1% Pro. Source data are provided as a Source Data file.

transitions of PLCDs. The stickers-and-spacers model for PSCP[41,50–52] has provided an organizing framework for uncovering sequence-specific driving forces of PLCDs[53–55]. The emerging picture may be summarized as follows: Aromatic residues function as stickers or cohesive motifs[53,54]. Physical crosslinks among stickers contribute to the networking of PLCDs within condensates[55]. The number of aromatic stickers is a key determinant of the threshold concentration for phase separation[53,54]. Further, across homologs, stickers are non-randomly and uniformly distributed along linear sequences of PLCDs[53,56]. Polar residues function as spacers interspersed between stickers. However, the identities and strengths of stickers are context-dependent and spacers are far from being inert entities[54]. Instead, the effective solvation volumes of spacers dictate the extent of coupling between networking, via reversible sticker-sticker crosslinks, and the driving forces for phase separation, via the influence on overall protein solubility[57]. For example, increasing the net charge of spacers can have a destabilizing effect on phase separation of PLCDs. Spacers also influence the cooperativity of sticker-sticker interactions[54].

Two recent studies generalized the binary stickers-and-spacers model[53], showing that Tyr is a stronger sticker than Phe within PLCDs[54,55]. Within PLCDs, Arg plays the role of an auxiliary sticker rather than a primary sticker. Further, the role of Arg as an auxiliary sticker is context-dependent, with some Arg residues in PLCDs playing the role of spacers rather than stickers. This stands in contrast to findings for full-length FET family proteins such as Fused in Sarcoma (FUS), where Tyr and Arg residues are the primary stickers in proteins that encompass both a PLCD and disordered RNA binding domains[52,58]. Accordingly, instead of being immutable[59], the relative strengths of sticker-sticker interactions and the identities of

primary versus auxiliary stickers are context-specific[39,60]. Context dependence and compositional specificity applies to spacers as well[52]. For example, Gly and Ser are non-equivalent as spacers in PLCDs[54,55]. For a given composition of stickers, Gly residues act as spacers that enhance the driving forces for phase separation when compared to Ser and other polar amino acids[54,55]. However, long poly-Gly tracts are generally avoided within PLCDs because these tracts become alternative stickers that drive fibril formation[61]. Evolutionarily, one observes correlations between sticker versus spacer identities in PLCDs. For example, the fractions of Tyr and Phe residues as well as Gly and Ser residues tend to be inversely correlated with one another. Additionally, positive correlations have been observed between Tyr and Gly contents as well as Phe and Ser contents[54]. These findings highlight the fact that sequence-encoded driving forces for phase separation of PLCDs and other IDRs are under evolutionary selection.

The driving forces for phase transitions of individual PLCDs depend on sequence-specific compositional biases[54,55]. Indeed, evolutionarily observed compositional variations can give rise to differences in driving forces for PSCP that can span several orders of magnitude[54]. Condensates such as stress granules and P bodies, and nuclear bodies such as nucleoli house ~40 or more known proteins with distinct PLCDs (Fig. 1b). Within a condensate, the PLCDs can be quite different from one another. The sequence lengths of PLCDs can vary by a factor of four, with most PLCDs being ~150 residues long (Fig. 1c). The driving forces for condensate formation in mixtures of PLCDs are of direct relevance for understanding how different PLCDs work together or in opposition to influence condensate formation, internal organization, and interfacial properties.

Here, we report results from studies of condensate formation in mixtures of PLCDs from two proteins, hnRNPA1[62] and FUS[63]. These two proteins are components of stress granules, paraspeckles, and other condensates[64]. Hereafter, we refer to the two PLCDs as A1-LCD and FUS-LCD, respectively. Mutations within both PLCDs are associated with the formation of aberrant stress granules in the context of Amyotrophic Lateral Sclerosis (ALS)[20,62,65]. The compositional differences between human versions of the two PLCDs are summarized in Fig. 1d. These differences translate to a positive net charge per residue (NCPR) for A1-LCD and a negative NCPR for FUS-LCD. In addition to being compositionally different, the two sequences have different lengths, with FUS-LCD being ~1.6 times longer than A1-LCD. However, despite being shorter than FUS-LCD, the driving forces for PSCP are stronger for A1-LCD when compared to FUS-LCD[55]. This is gleaned from comparative measurements of the temperature- and solution-condition-dependent saturation concentrations ($c_{sat}$) for phase separation. Here, we focus on understanding how the interplay of homotypic and heterotypic interactions influences the driving forces for condensate formation in mixtures of A1-LCD and FUS-LCD molecules. For this, we deployed a recently developed coarse-grained model to simulate temperature-dependent phase transitions of mixtures of A1-LCD and FUS-LCD[55]. In the model, the details of which have been previously published[55], each PLCD residue is modeled as a single bead with distinct inter-bead interactions.

The simulations were performed using LaSSI[51], which is a lattice-based Monte Carlo simulation engine. The bespoke coarse-grained model of Farag et al.[55], is transferrable unto PLCDs of similar compositional biases, but, as detailed by Farag et al., it is not transferable unto IDRs that are not PLCDs. Instead, the Gaussian process Bayesian optimization machine learning paradigm[66] for developing bespoke coarse-grained models is transferrable to other systems[51,55]. The lattice model is based on the generalized bond fluctuation model[67,68], and this allows for the incorporation of anisotropic interactions as described in the original work of Choi et al.[51]. The transferability of the parameterization paradigm to distinct families of IDRs, creates a basis set of Hamiltonians, allowing for understanding IDR phase behaviors as flows through the space of Hamiltonians[69] that define distinct families. These considerations will be taken up elsewhere. Here, we focus on insights that emerge from the computations, the predictions from which are tested using experiments, for mixtures of PLCDs.

We find that heterotypic interactions are the dominant contributors to condensate formation in 1:1 mixtures of A1-LCD and FUS-LCD. This leads to two-component phase diagrams with distinctive shapes and slopes for tie lines. We test the accuracy of our computational predictions using in vitro experiments that leverage an analytical HPLC-based method for measuring phase diagrams in multicomponent mixtures[70]. Further, through additional simulations, we uncover general rules for how the interplay between homotypic and heterotypic interactions influences the internal organization of distinct molecules and interfacial properties of multicomponent condensates.

## Results

### Heterotyhpic interactions enhance the driving forces for phase separation of mixtures of A1-LCD and FUS-LCD

We performed a series of simulations for mixtures of A1-LCD and FUS-LCD. The sequences of A1-LCD and FUS-LCD are shown in Fig. 2a. The total protein concentrations were fixed in the simulations, and the ratios of FUS-LCD-to-A1-LCD were varied from one set of simulations to another. Treating the mixture as a system with one type of macromolecule, we computed binodals in the plane of total protein concentration along the abscissa and temperature along the ordinate. This approach to depicting phase boundaries for mixtures parallels that of Elbaum-Garfinkle et al.[71], and Wei et al.[72]. Based on the computed binodals (Fig. 2b) we predict that a 1:1 mixture of FUS-LCD and A1-LCD

undergoes phase separation at a lower total protein concentration than either FUS-LCD or A1-LCD on its own. This is suggestive of the presence of heterotypic interactions that enhance phase separation.

To better understand the interplay between homotypic and heterotypic interactions, we recast the results as phase boundaries in a two-parameter space at a fixed temperature, where each axis is defined by the concentration of one protein. In Fig. 2c we show expectations for the dilute arms of two-dimensional phase boundaries for a system of two components that undergoes co-phase separation from solution.

At the temperature of interest, we shall denote the saturation concentration of A1-LCD in the absence of FUS-LCD as $c_{sat,A1}$. Likewise, at the same simulation temperature, the saturation concentration of FUS-LCD in the absence of A1-LCD is denoted as $c_{sat,FUS}$. If the dilute arm of the two-dimensional phase boundary is a straight line joining the individual $c_{sat}$ values, then the contributions to phase separation of the mixture of PLCDs are purely additive, and the co-condensate is a random mixture. In this case, homotypic and heterotypic interactions make equivalent contributions to the driving forces for phase separation. As a result, the total protein concentration in the dilute phase of the two-phase system can be written as: $c_{dilute} = ac_{sat,A1} + (1 - a)c_{sat,FUS}$, where $a$ is the fraction of A1-LCD molecules in the system; when $a = 1$, $c_{dilute} = c_{sat,A1}$, and when $a = 0$, $c_{dilute} = c_{sat,FUS}$. This situation can be visualized by simulating a pseudo-binary mixture comprising A1-LCD molecules where half of the molecules are labeled as type A and the other half is labeled as type B. In this case, the heterotypic and homotypic interactions among type A and B molecules are all equivalent, resulting in a straight line for the dilute arm of the phase diagram (Supplementary Fig. 1). If the computed or measured dilute arm of the two-dimensional phase boundary is concave, then heterotypic interactions enhance the driving forces for phase separation. Conversely, if the dilute arm of the two-dimensional phase diagram is convex, then heterotypic interactions weaken the driving forces for phase separation. Importantly, the heterotypic interactions do not need to be repulsive for the phase boundary to have a convex shape. Rather, they only need to be less attractive than the homotypic interactions while being competitive enough to weaken the latter. Furthermore, the degree to which the driving forces for phase separation are enhanced or weakened will depend on the stoichiometric ratio of the two components. A concise treatment of some of the nuances underlying mathematical aspects of shapes of phase boundaries that result from the interplay between homotypic and heterotypic interactions and its effect on buffering has been described by Deviri and Safran[73]. Their work provides a useful introduction to the concepts that drive our analysis, and it was based in part on the original work of Choi et al.[51], Riback et al.[45], and Seim et al.[28].

The results shown in Fig. 2b were recast by fixing the simulation temperature and plotting the computed dilute phase concentrations of A1-LCD and FUS-LCD for different stoichiometric ratios (Fig. 2d). We observe a concave shape, indicating an enhancement of phase separation via heterotypic interactions. We sought the simplest, most plausible explanation for our observations. We note that FUS-LCD is negatively charged, and A1-LCD is positively charged. Accordingly, we reasoned that complementary electrostatic interactions are likely to enhance the driving forces for co-phase separation. These interactions are likely to contribute in addition to heterotypic aromatic sticker interactions. To test this hypothesis, we performed simulations for mixtures of FUS-LCD and a variant of A1-LCD denoted as A1-LCD+12D[54]. In this variant, twelve Asp residues were substituted across the sequence, replacing extant spacers, giving rise to a sequence with a net negative charge. Although the aromatic sticker interactions remain unchanged, the electrostatic interactions should be weakened. We reasoned that this would generate a dilute arm with a more convex shape, and this is precisely what we observe (Fig. 2e). Similarly, we performed simulations of a positively charged variant of FUS-LCD that was prescribed to have the same net charge per residue as A1-LCD,

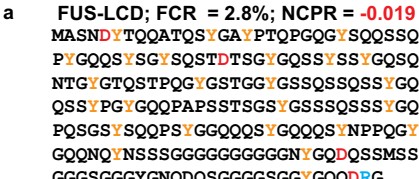

**a**

**FUS-LCD; FCR = 2.8%; NCPR = -0.019**

MASNDYTQQATQSYGAYPTQPGQGYSQQSSQ
PYGQQSYSGYSQSTDTSGYGQSSYSSYGQSQ
NTGYGTQSTPQGYGSTGGYGSSQSSQSSYGQ
QSSYPGYGQQPAPSSTSGSYGSSSQSSSYGQ
PQSGSYSQQPSYGQQQSYGQQQSYNPPQGY
GQQNQYNSSSGGGGGGGGGGGNYGQDQSSMSS
GGGGSGGGYGNQDQSGGGGSGGYGQQDRG

**A1-LCD; FCR = 11.7%; NCPR = 0.058**

MASASSSQRGRSGSGNFGGGRGGGFGGNDNF
GRGGNFSGRGGFGGSRGGGGYGGSGDGYNGF
GNDGSNFGGGGSYNDFGNYNNQSSNFGPMKG
GNFGGRSSGGSGGGGQYFAKPRNQGGYGGSS
SSSSYGSGRRF

**A1-LCD +12D; FCR = 20.4%; NCPR = -0.029**

MASADSSQRDRDDSGNFGDGRGGGFGGNDNF
GRGGNFSDRGGFGGSRGDGGYGGDGDGYNGF
GNDGSNFGGGGSYNDFGNYNNQSSNFDPMKG
GNFGDRSSGPYDGGGQYFAKPRNQGGYGGSS
SSSSYGSDRRF

**Fig. 2 | LaSSI simulations predict that phase behaviors of mixtures of PLCDs are governed by the interplay of homotypic and heterotypic interactions. a** Details of the amino acid sequences of FUS-LCD, A1-LCD, and A1-LCD + 12D, including the fraction of charged residues (FCR) and the net charge per residue (NCPR). FUS-LCD has 214 residues, whereas A1-LCD and A1-LCD + 12D each have 135 residues. **b** Computed binodals of mixtures of FUS-LCD and A1-LCD. Volume fractions correspond to the sum of the FUS-LCD and A1-LCD volume fractions. Note that volume fractions are unitless quantities. Percentages described in the legend indicate the relative percentage of FUS-LCD molecules in the binary mixture. **c** Schematic depicting the expected shape of the low concentration arm of the binodal depending on the interplay between homotypic vs. heterotypic interactions. Computed, low-concentration arms of binodals for mixtures of FUS-LCD and **(d)** A1-LCD at a fixed simulation temperature of 50 or **(e)** A1-LCD + 12D at a fixed simulation temperature of 50. Note that these are reduced temperatures, calibrated in terms of $RT$, where $R = 1$ contact energy unit. Previous work showed how these simulation temperatures can be converted to temperatures in a test tube[53,55]. We avoid that here because the conversions were not calibrated for mixtures of PLCDs. Black dashed lines connect the intrinsic $c_{sat}$ of FUS-LCD to the intrinsic $c_{sat}$ of A1-LCD or A1-LCD + 12D and are shown to indicate the expected shape of the low concentration arm of the binodal if heterotypic interactions are on par with homotypic interactions. n = 10 independent simulations with random starting seeds. Error bars in **(b), (d)**, and **(e)** are standard errors about the mean. Where error bars are invisible, they are smaller than the marker size. Source data are provided as a Source Data file.

which we refer to as FUS-LCD (+). Again, the dilute arm shows a convex shape, suggesting that the heterotypic interactions are weakened relative to the mixture of FUS-LCD and A1-LCD (Supplementary Fig. 2).

## Results from in vitro measurements are in accord with computational predictions

Next, we measured co-phase separation in aqueous mixtures of A1-LCD and FUS-LCD molecules (see Materials and Methods, *Supplementary Material* and Supplementary Fig. 3). Diffraction-limited fluorescence microscopy shows that the PLCDs co-localize into the same condensates upon phase separation (Fig. 3a). This is true for all

concentration ratios studied. We then used our recently described analytical high-performance liquid chromatography (HPLC) method[70] to determine dilute and dense phase concentrations of both species in mixtures with different mass concentration ratios of the two PLCDs.

As in the simulations, a 1:1 mixture of A1-LCD and FUS-LCD undergoes phase separation at a lower concentration than either protein on its own (Fig. 3b). Recasting the results in terms of a two-dimensional phase boundary shows the same patterns as the simulations, where the dilute arm of the phase boundary has a concave shape (Fig. 3c). These results suggest that complementary electrostatic interactions contribute to enhance co-phase separation of FUS-LCD

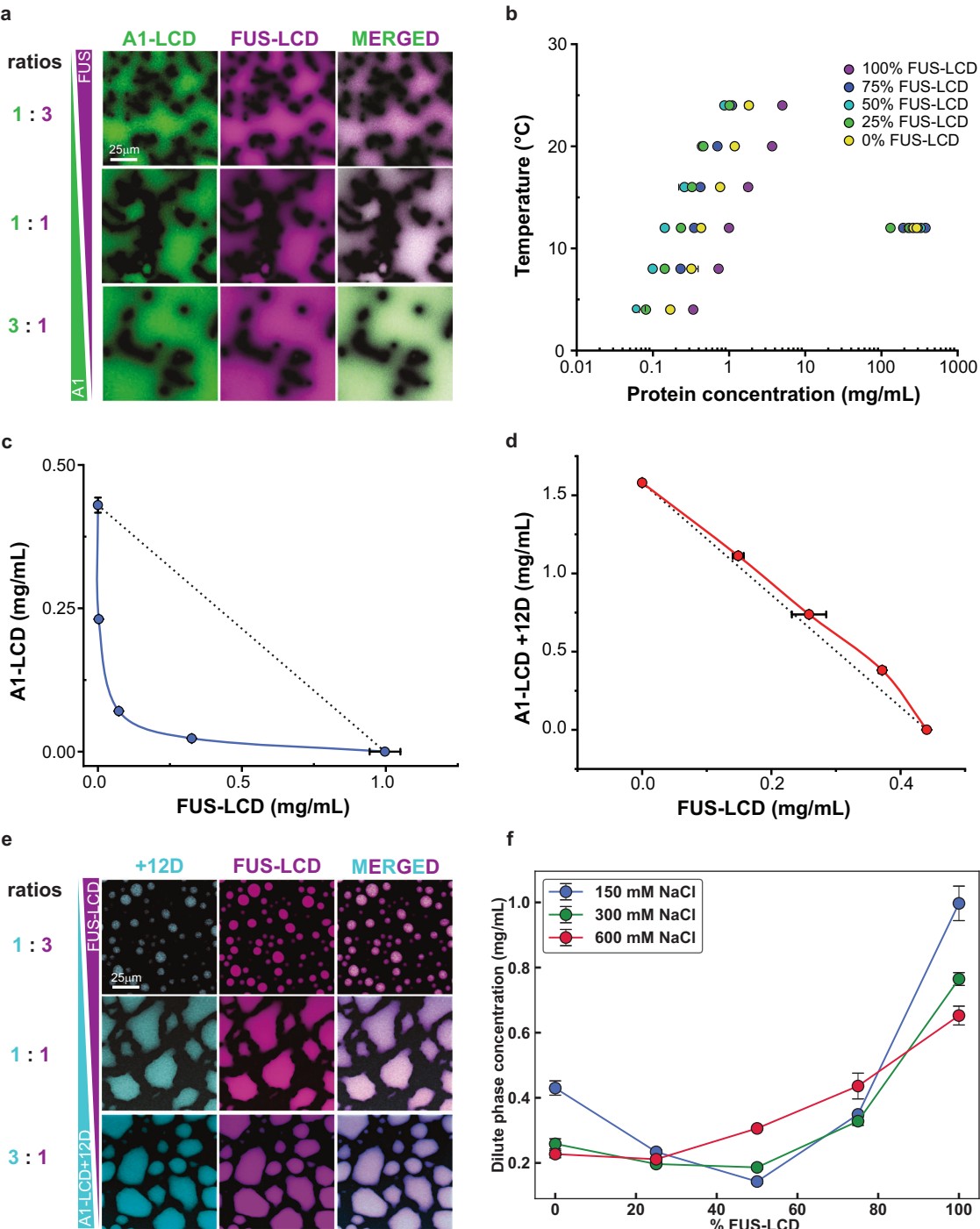

**Fig. 3 | Data from in vitro phase separation assays of mixtures of PLCDs agree with simulation predictions. a** Confocal fluorescence microscopy images of mixtures of FUS-LCD and A1-LCD at various mass concentration ratios. **b** Measured binodals of mixtures of FUS-LCD and A1-LCD. Concentrations correspond to the sum of the FUS-LCD and A1-LCD concentrations. Percentages in the legend indicate the mass concentration ratio of FUS-LCD to A1-LCD. Measured 2-component binodals of mixtures of FUS-LCD and (**c**) A1-LCD at 12 °C or (**d**) A1-LCD + 12D at 4 °C. Black dashed lines connect the intrinsic $c_{sat}$ of FUS-LCD to the intrinsic $c_{sat}$ of A1-LCD or A1-LCD + 12D and are shown to indicate the expected binodal shape if heterotypic interactions are on par with homotypic interactions. **e** Confocal fluorescence microscopy images of mixtures of FUS-LCD and A1-LCD + 12D at various mass concentration ratios. **f** Measured dilute phase total protein concentrations of mixtures of FUS-LCD and A1-LCD at various mass concentration ratios. Data are shown for mixtures in 150 mM NaCl, 300 mM NaCl, or 600 mM NaCl. $n = 3$ independent experiments for all phase separation data. For microscopy experiments, three images were analyzed at each condition. Error bars indicate standard deviations about the mean. Source data are provided as a Source Data file.

and A1-LCD. Note that there are differences in the extent of concavity observed in experiments. This highlights two sources of weakness with the computations: First, the treatment of electrostatic interactions use a mean-field model that is based on the NCPR values. Second, ion effects such as differential partitioning into dense versus dilute phases are not accounted for in the simulations. However, the trends are similar across the two modes of investigation, and this point is underscored in comparisons of phase behaviors in mixtures of the FUS-LCD and A1-LCD+12D system. In contrast, and in agreement with computational predictions, the dilute arm of the measured phase

diagram for mixtures of the FUS-LCD and A1-LCD+12D system has a more convex shape (Fig. 3d), which is due to a lack of complementary electrostatic interactions.

Notably, although mixtures of FUS-LCD and A1-LCD+12D have a weakened driving force for phase separation relative to FUS-LCD and A1-LCD, the two proteins still co-localize into a single dense phase as gleaned from diffraction-limited microscopy (Fig. 3e). We measured how the total dilute phase protein concentration varied as a function of salt concentration and the ratio of FUS-LCD-to-A1-LCD (Fig. 3f). Increasing the salt concentration causes a decrease in $c_{sat}$ values for either protein on its own, but an increased $c_{dilute}$ value for a 1:1 mixture. Note that $c_{dilute}$ for the mixture is the sum of dilute phase concentrations of each of the proteins. The curve connecting the dilute phase concentrations has a significantly smaller curvature. We rationalize this as follows: In solutions with only one type of PLCD, all proteins have the same sign and magnitude of charge. Accordingly, the intermolecular electrostatic interactions will be repulsive. Increasing the salt concentration screens repulsive interactions, thereby lowering the intrinsic $c_{sat}$ values. In contrast, the 1:1 mixture contains both repulsive homotypic electrostatic interactions and attractive heterotypic electrostatic interactions. By increasing the salt concentration, both types of interactions are weakened. Because the heterotypic interactions are stronger than the homotypic interactions at the lowest salt concentration studied, this results in a net loss of attractive interactions and an increase in $c_{dilute}$. More importantly, the fact that the curvature decreases indicates that the gap between homotypic and heterotypic interactions narrows as the salt concentration is increased. These results suggest an interplay between homotypic and heterotypic aromatic as well as electrostatic interactions in mixtures of FUS- and A1-LCD.

## Tie lines and their slopes help uncover the relative contributions of homotypic versus heterotypic interactions to phase separation in mixtures of PLCDs

In a system that forms precisely two coexisting phases, the generalized tie simplex[39] for a given composition will be a tie line. This is a straight line that connects points of equal chemical potentials and osmotic pressures on the dilute and dense arms of phase boundaries[74]. The signs and magnitudes of the slopes of tie lines provide quantitative insights regarding the interplay between homotypic versus heterotypic interactions[74].

In a system with macromolecules A and B that undergo co-phase separation from a solvent, we can map the low and high concentration arms of phase boundaries on a plane with the concentration of A being the variable along the abscissa, and the concentration of B being the variable along the ordinate (Fig. 4). In these titrations, we fix the solution conditions including the temperature. The slope of the tie line will be unity if heterotypic interactions are the dominant drivers of phase separation or if the homotypic interactions are equivalent to each other while also being on par with the heterotypic interactions. In the A-B mixture, a tie line with a slope that is less than unity will imply that the concentration of component A changes more significantly across the phase boundary than that of component B. This would suggest that component A, specifically homotypic interactions of component A, dominate as drivers of phase separation. Conversely, if tie lines have slopes that are greater than unity, then the homotypic interactions among B molecules are the stronger drivers of phase separation. Note that these pronouncements regarding numerical values of slopes of tie lines are governed by the choices of the axes we assign to concentrations of components A and B.

If the total concentration of the mixture is $c_{tot}$, we can extract tie lines by joining three points corresponding to $c_{dilute}$, $c_{tot}$, and $c_{dense}$. Here, $c_{dilute}$ and $c_{dense}$ are the macromolecular concentrations in the coexisting dilute and dense phases, respectively. If phase separation gives rise to precisely two coexisting phases, then a single line should

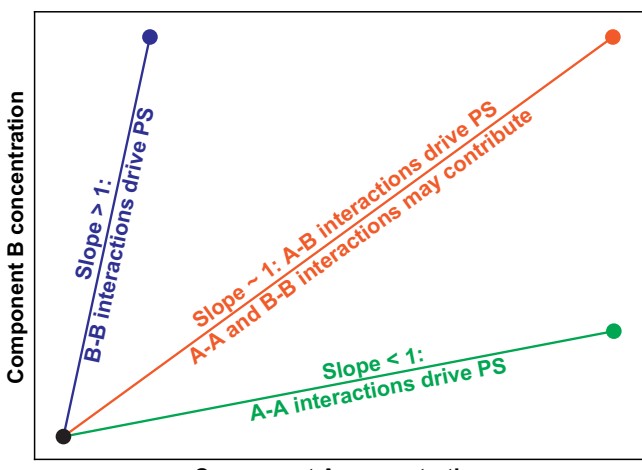

**Fig. 4 | Schematic depicting how to interpret tie lines for a system of two components that undergo co-phase separation from the solvent to form precisely two coexisting phases.** Concentrations of the two components, A and B, are titrated along the abscissa and ordinate, respectively. The A-B mixture undergoes phase separation to form two coexisting phases. If homotypic, A-A interactions are the most dominant, then the slope of the tie line that connects the concentrations of the two coexisting phases will be less than unity (green line). Likewise, if homotypic B-B interactions are most dominant, then the slope of the tie line that connects the concentrations of the two coexisting phases will be greater than unity (blue line). If heterotypic A-B interactions and homotypic A-A as well as B-B interactions make equivalent contributions to phase separation, then the slope of the tie line will be ≈ 1 (red line).

connect the three points. To test that this is the case, we plotted two sets of lines for each mixture *viz.*, one that joins $c_{dilute}$ and $c_{tot}$ and another that joins $c_{tot}$ and $c_{dense}$. If the slopes of the two lines are identical or nearly identical, then the system forms two coexisting phases upon phase separation, and the hidden components do not have major effects on the overall phase behavior[75]. The tie lines computed in this manner are shown in Figs. 5a and 5b for simulations of different ratios of A1-LCD-to-FUS-LCD. The slopes of the two sets of tie lines are essentially equivalent to one another (Fig. 5c). We repeated this analysis for measured phase boundaries of the corresponding system in vitro and find similar values for the slopes of the tie lines (Fig. 5d–f).

In computations and measurements, the 1:1 mixtures of A1-LCD and FUS-LCD have tie lines with slopes that are essentially one. Therefore, phase separation involves changes in concentrations from the dilute to the dense phase that are similar for both sets of molecules. This suggests that heterotypic interactions are the dominant drivers of phase separation in 1:1 mixtures. Conversely, in a mixture with a 3:1 ratio of A1-LCD-to-FUS-LCD, the slope of the tie line is significantly greater than one. The implication is that homotypic interactions among A1-LCD molecules play a dominant role in driving phase separation along this tie line. Likewise, in a mixture with a 1:3 ratio of A1-LCD-to-FUS-LCD molecules, the slope of the tie line is significantly less than one, highlighting the importance of homotypic interactions among FUS-LCD molecules along this tie line.

Next, we asked if the tie lines for the mixture of FUS-LCD and A1-LCD+12D would differ from those of FUS-LCD and A1-LCD. The intrinsic $c_{sat}$ of FUS-LCD is approximately four times lower than that of A1-LCD+12D. First, we plotted the low-concentration arms obtained from in vitro measurements (Fig. 6a). Following Qian et al[74,75], the agreement between the two sets of tie lines in Fig. 5f suggests that we can infer the slope of the whole tie line based on the slope of the line that connects $c_{dilute}$ to $c_{tot}$. Unlike the mixture of A1-LCD and FUS-LCD, we find that for the FUS-LCD and A1-LCD+12D

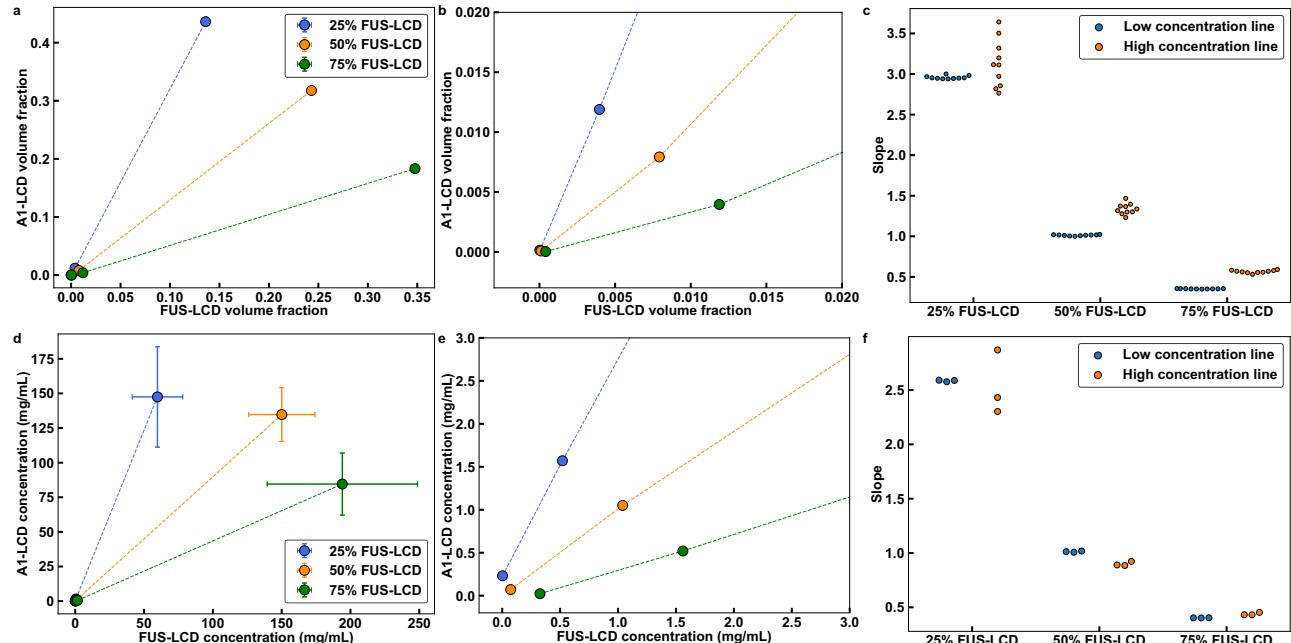

**Fig. 5 | Tie lines of two-dimensional phase diagrams provide information regarding the interplay between heterotypic and homotypic interactions.** Here, results shown in panels (**a**–**c**) are from simulations and those in panels (**d**–**f**) are from in vitro measurements. (**a**), (**b**) Tie lines extracted from simulations of phase separation in mixtures of FUS-LCD and A1-LCD. The simulation temperature is 50 in reduced units (see *Supplementary Material*). **a** This plot includes the dilute phase concentrations, the total concentrations, and the dense phase concentrations. **b** This plot shows only the dilute phase concentrations and the total concentrations. **c** Computed slopes of the tie lines in **b** connecting dilute phase concentrations to the total concentrations and the total concentrations to the dense phase concentrations. The values are very similar to one another. **d, e** Tie lines extracted from in vitro measurements of the FUS-LCD and A1-LCD system obtained at a temperature of 12 °C. **d** Plot shows the dilute phase concentrations, the total concentrations, and the dense phase concentrations, whereas (**e**) focuses on the dilute phase concentrations and the total concentrations. **f** The slopes of the tie lines in (**e**) connecting dilute phase concentrations to the total concentrations and the total concentrations to the dense phase concentrations. $n = 10$ independent simulations with random starting seeds or 3 independent experiments. Error bars in all panels are standard deviations about the mean. Total system concentrations do not have error bars. Otherwise, where error bars are invisible, they are smaller than the marker size. Source data are provided as a Source Data file.

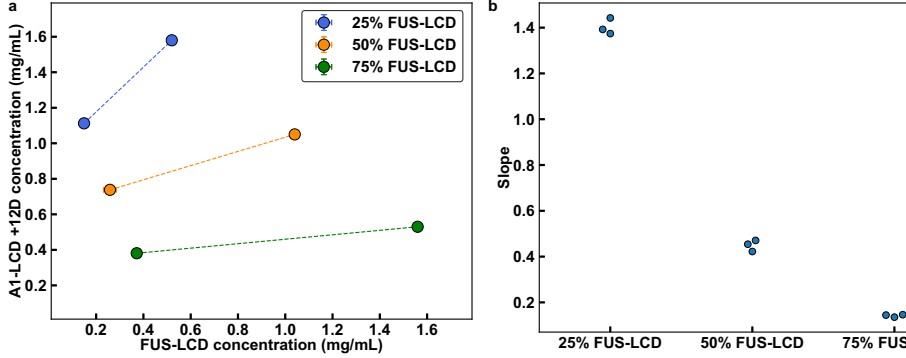

**Fig. 6 | Weakening heterotypic interactions causes tie lines to be more sensitive to homotypic interactions. a** Tie lines of the in vitro FUS-LCD and A1-LCD + 12D system at a temperature of 4 °C. These tie lines connect dilute phase concentrations to total concentrations. **b** Tie line slopes from (**a**). $n = 3$ independent experiments. Error bars in all panels are standard deviations about the mean. Total system concentrations do not have associated error bars. Otherwise, where error bars are invisible, they are smaller than the marker size. Source data are provided as a Source Data file.

mixture, the slope of the tie line for the 1:1 mixture is significantly less than one (Fig. 6b). This suggests that along this tie line, the homotypic interactions among FUS-LCD interactions are major drivers of phase separation. This effect is further enhanced for the mixture with a 3:1 ratio of FUS-LCD-to-A1-LCD+12D. Finally, for a mixture with a 1:3 ratio of FUS-LCD-to-A1-LCD+12D, the slope of the tie line is closer to unity, indicating that along this tie line, the interplay among heterotypic interactions, homotypic FUS-LCD interactions, and homotypic A1-LCD+12D interactions, are relatively well-balanced (Fig. 6b). Taken together, we find that if heterotypic interactions are not the dominant drivers of phase separation, then co-phase separation is driven by molecules that have stronger intrinsic driving forces for phase separation. We also performed this analysis for the simulations of mixtures of FUS-LCD and A1-LCD+12D (Supplementary Fig. 4). Although our model accurately predicts the intrinsic $c_{sat}$ of each individual component within a factor of 2.5, it inaccurately predicts that the intrinsic $c_{sat}$ values of the two proteins differ by less than a factor of two. Because of this, the slopes of the tie lines gleaned from the simulations indicate that the dominant driver of phase separation is the protein with a higher concentration in solution.

## How condensate interfaces are influenced by the balance of homotypic versus heterotypic interactions

Next, we followed recent approaches[55] and computed radial density distributions, and logistic fits to these distributions[76] for a 1:1 mixture of A1-LCD and FUS-LCD (Fig. 7a). We find that the concentration of A1-LCD is greater than that of FUS-LCD in the dense phase and less than that of FUS-LCD in the dilute phase. This is because the intrinsic $c_{sat}$ of A1-LCD is less than that of FUS-LCD. The thickness of the interface as predicted by the logistic fit to the radial density distribution for FUS-LCD is larger than that of A1-LCD. This is because FUS-LCD is ~1.6 times longer than A1-LCD.

We analyzed the ensemble averaged radius of gyration, $R_g$, of each species, normalized by $\sqrt{N}$, where $N$ is the protein length (Fig. 7b). In the dense phase, each species has a similar value of $\frac{R_g}{\sqrt{N}}$, because the dense phase is a better solvent for both species when compared to the dilute phase, where the solvent is relatively poor for both species[55]. In the dilute phase, $\frac{R_g}{\sqrt{N}}$ is slightly larger for FUS-LCD than for A1-LCD, and this is in accordance with the weaker homotypic interactions among FUS-LCD molecules. At the interface, both species show the predicted chain expansion[55].

Next, we probed the internal organization of A1-LCD and FUS-LCD molecules with respect to one another. To quantify this, we deployed a crosslinking parameter $L_{i \cdot j}$ to determine the relative likelihood that protein species $i$ interacts with species $j$, given a fixed total number of proteins from each species in the condensate. Details of how $L_{i \cdot j}$ is computed are described in the *Supplementary Material*.

Values of $L_{i \cdot j}$ that are close to one indicate that species $i$ interacts with species $j$ as they would in a random mixture, being proportional to the number of $i$ and $j$ proteins in the condensate. In contrast, values of $L_{i \cdot j}$ that are greater than or less than one point to a non-random organization of different molecules within the condensate. If $L_{i \cdot j} > 1$ then species $i$ is more likely to interact with species $j$ than would be expected from random mixing. Conversely, if $L_{i \cdot j} < 1$, then species $i$ is less likely to interact with species $j$ than would be expected for a random mixture.

We calculated $L_{i \cdot j}$ for simulations of FUS-LCD and A1-LCD at 1:1 ratios and a series of temperatures that correspond to the two-phase regime. We found the following trends: In general, FUS-LCD molecules are more likely to interact with A1-LCD molecules than with other FUS-LCD molecules. In contrast, A1-LCD molecules show relatively equal preferences for interacting with FUS-LCD or A1-LCD molecules. As the temperature increases, both FUS-LCD and A1-LCD molecules show an increased preference for interacting with A1-LCD molecules. This is because the width of the two-phase regime narrows more rapidly for FUS-LCD than for A1-LCD[55]. The results outlined in Fig. 7 set up predictions for internal organizations and interfacial properties based on protein length and the interplay between homotypic and heterotypic interactions.

## Rules for internal and interfacial organization in mixtures that form co-localized condensates

To uncover general rules for how the interplay of homotypic versus heterotypic interactions influences the internal organizations of condensates, we chose simpler systems and investigated these in a series of separate simulations. These homopolymeric systems have been shown to be useful proxies of PLCDs[54,55]. We performed simulations of 1:1 mixtures of two homopolymers, designated as A and B. Each polymer has 150 beads, and we varied the strengths of homotypic (A-A and B-B interactions) and heterotypic (A-B interaction) energies. For each model, we calculated $L_{i \cdot j}$ at various temperatures corresponding to the two-phase regime of the composite system.

First, we set all homotypic and heterotypic interactions to be equivalent. In this scenario, we find that $L_{i \cdot j}$ equals one for all species

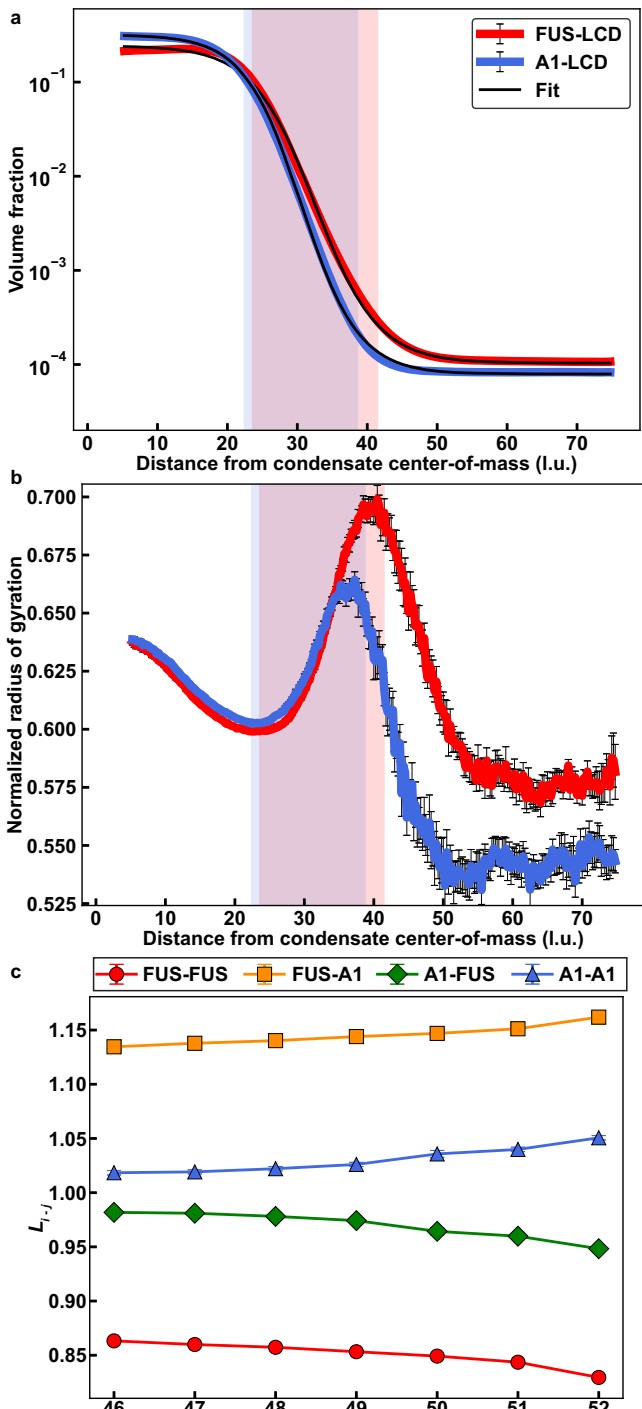

**Fig. 7 | Simulated mixtures of PLCDs display complex interfacial features and internal organizations. a** Radial density plots of simulated FUS-LCD and A1-LCD in a 1:1 mixture. Solid black curves indicate logistic fits to each radial density profile. **b** Radius of gyration normalized by $\sqrt{N}$, where $N$ is the chain length, as a function of the distance from the condensate center-of-mass. **c** Values of $L_{i \cdot j}$ (see text) as a function of simulation temperature for a 1:1 mass concentration mixture of FUS-LCD and A1-LCD. In **a** and **b**, simulations were performed at a simulation temperature of 49, and translucent rectangles indicate the interfacial regions predicted by the corresponding logistic fit of FUS-LCD (red) and A1-LCD (blue). $n = 10$ independent simulations with random starting seeds. Error bars in all panels are standard errors about the mean. Source data are provided as a Source Data file.

combinations and at all temperatures, suggesting A and B are randomly mixed within the condensate (Fig. 8a). However, if heterotypic interactions are stronger than homotypic interactions, then we observe an increase in A-B contacts, indicating that chains within the condensate are organized to maximize heterotypic interactions (Fig. 8b). Setting the homotypic interactions to be stronger than heterotypic interactions gives rise to internally demixed condensates with an A-rich region forming an interface with a B-rich region (Fig. 8c). Similar findings were recently reported by Welles et al.[77]. Finally, adding asymmetry to the energetics by keeping A-B and B-B interactions equivalent, but weakening A-A interactions leads to a strong preference for A molecules to interact with B molecules over other A molecules, whereas B molecules show indifference at low temperatures, but a preference for other B molecules at high temperatures (Fig. 8d). These observations are akin to those made for mixtures of A1-LCD and FUS-LCD molecules (Fig. 7c).

We also determined the effects of changing the stoichiometric ratio of A and B on the internal organizations of the systems described in Fig. 8. When all interactions are equivalent, the condensate is still a random mixture of A and B regardless of the relative concentrations of the two polymers (Supplementary Fig. 5a). When heterotypic interactions are stronger than homotypic interactions, there is still a strong preference for polymers to interact with molecules of the opposite type, and an asymmetry emerges across the two types of polymers (Supplementary Fig. 5b). To illustrate this, we consider the 3:1 mixture of A:B where we find $L_{A-B}$ to be greater than $L_{B-A}$. This is because heterotypic crosslinks are less likely to occur at random from the perspective of an A molecule than from the perspective of a B molecule since an A molecule has fewer heterotypic partners with which to interact. A similar asymmetry occurs when homotypic interactions outweigh heterotypic interactions (Supplementary Fig. 5c). As in the equal concentration mixture, we still see the emergence of an A-rich region interfacing with a B-rich region. In the 3:1 mixture of A:B, we now find that $L_{B-B}$ is significantly greater than $L_{A-A}$. This is because both types of polymers would be randomly expected to interact with A molecules more often than with B molecules. Higher likelihoods of B-B interactions cannot occur at random given the concentrations of molecules in the condensate. Lastly, keeping A-B and B-B interactions equivalent, but weakening A-A interactions still leads to the same overall trends as in Fig. 8d (Supplementary Fig. 5d). However, the crossover temperature where B molecules show a stronger preference for B-B interactions as opposed to B-A interactions is highly dependent on the system stoichiometry. As the ratio of A:B molecules decreases, this crossover temperature increases due to the increased number of B molecules in the system increasing the probability of a B-B crosslink at random. Taken together, these results highlight how diverse molecular stoichiometries and interaction hierarchies can result in decidedly nonrandom internal organizations within condensates.

The lengths of macromolecules will influence the material properties[78] and interfacial features of condensates[55]. Therefore, we asked how sequence length affects internal and interfacial features of condensates using simulations of homopolymers of various lengths. We performed simulations with equivalent mass concentrations for mixtures of homopolymers of lengths 150 (H150) and 300 (H300) where all homotypic and heterotypic interactions were set to be equal (Fig. 9a). Calculated phase diagrams show that H300 has a lower $c_{sat}$ than H150 (*Supplementary Material*, Supplementary Fig. 6a). In addition, analysis of $L_{i-j}$ for the mixture shows that there is a strong preference for interacting with H300 above $T \sim 56$ in reduced units. Beyond this temperature, H150 no longer phase separates on its own (Supplementary Fig. 6a). Radial density distributions show that H300 has a stronger preference for the dense phase and a weaker preference for the dilute phase when compared to H150. Logistic fits of the radial densities, used to determine interfacial regions corresponding to each homopolymer, show that the interfacial region is wider for H300 than

for H150. We also analyzed the normalized radius of gyration and found that H300 has a greater degree of chain expansion in the interface when compared to H150. This is true even though it is more compact than H150 in the dense and dilute phases. Along the radial coordinate from the center of the condensate, the location within the interface that corresponds to the peak of chain expansion of H300 is shifted closer to the dilute phase when compared to the corresponding peak for H150.

To assess the extent to which our findings reflect the stronger intrinsic driving forces for phase separation of longer homopolymers, we titrated the strengths of homotypic and heterotypic interactions of H300 and H150 so that the $c_{sat}$ values of H300 and H150 were equivalent (Fig. 9b and *Supplementary Material*, Supplementary Fig. 6b). In mixtures of equivalent mass concentrations, the mixture of H150 and H300 molecules forms apparent core-shell structures that allow both H150 and H300 to maximize their interactions with H150 molecules, as shown by an analysis of $L_{i-j}$ (*Supplementary Material*, Supplementary Fig. 6b). In turn, the width of the interfacial region defined by H300 shrinks when compared to the results shown in Fig. 9a. The scaled $R_g$ values for H150 and H300 are similar in the dilute and dense phases. However, the locations and heights of the peaks of chain expansion show the same pattern as in Fig. 9a.

Finally, we asked if interfacial features vary when we extend to three-component systems. In simulations that use equal mass concentrations of homopolymers of lengths 200, 300, and 400, with all interaction strengths being equal, we find that interfacial organizational features follow rules that were uncovered for binary mixtures. This is evident in comparisons of results shown in Fig. 9c and *Supplementary Material*, Supplementary Fig. 6c to those shown in Fig. 9a and *Supplementary Material*, Supplementary Fig. 6a. Longer polymers make up more of the outer regions of the interface. They also show a greater level of expansion at the interface, and the peak of chain expansion shifts toward the dilute phase relative to that of the shorter polymers. These results highlight how polydisperse mixtures of IDRs with similar compositional biases and different lengths can give rise to distinct internal organization and interfacial preferences.

## Discussion

Condensates are characterized by distinctive macromolecular compositions[79–83]. It is thought that macromolecular compositions, delineated into categories of molecules known as scaffolds, clients, regulators, crowders, and ligands, contribute to the functions of condensates[84–89]. Here, we explored how the interplay between homotypic versus heterotypic interactions of PLCDs influence the driving forces for phase separation. The FUS-LCD and A1-LCD are intrinsically disordered scaffolds with at least cursorily similar compositional biases. Our results set the stage for understanding how control over compositions in complex mixtures might influence condensate formation of bodies such as 40 S hnRNP particles[90] or stress granules. There is also growing interest in understanding how interplay of homotypic and heterotypic interactions impact the dynamically controlled[91] or purely thermodynamically influenced miscibility of condensates formed in multicomponent systems[77,92–94].

Key findings that emerge from our investigations are as follows: In 1:1 binary mixtures of PLCDs, heterotypic interactions can enhance the driving forces for phase separation. This enhancement comes, in part, from complementary electrostatic interactions, thus suggesting a role for complex coacervation-like processes[43,44,95] even in systems where fewer than 10% of the residues are charged. This highlights additional roles for spacer residues, which determine the solubility and the extent of coupling between associative and segregative transitions for individual PLCDs[39], while they determine the electrostatic complementarity of mixtures of PLCDs. Within condensates, the concentration of macromolecules will be above the overlap threshold[55]. As a result, the interplay between homotypic and heterotypic interactions will also

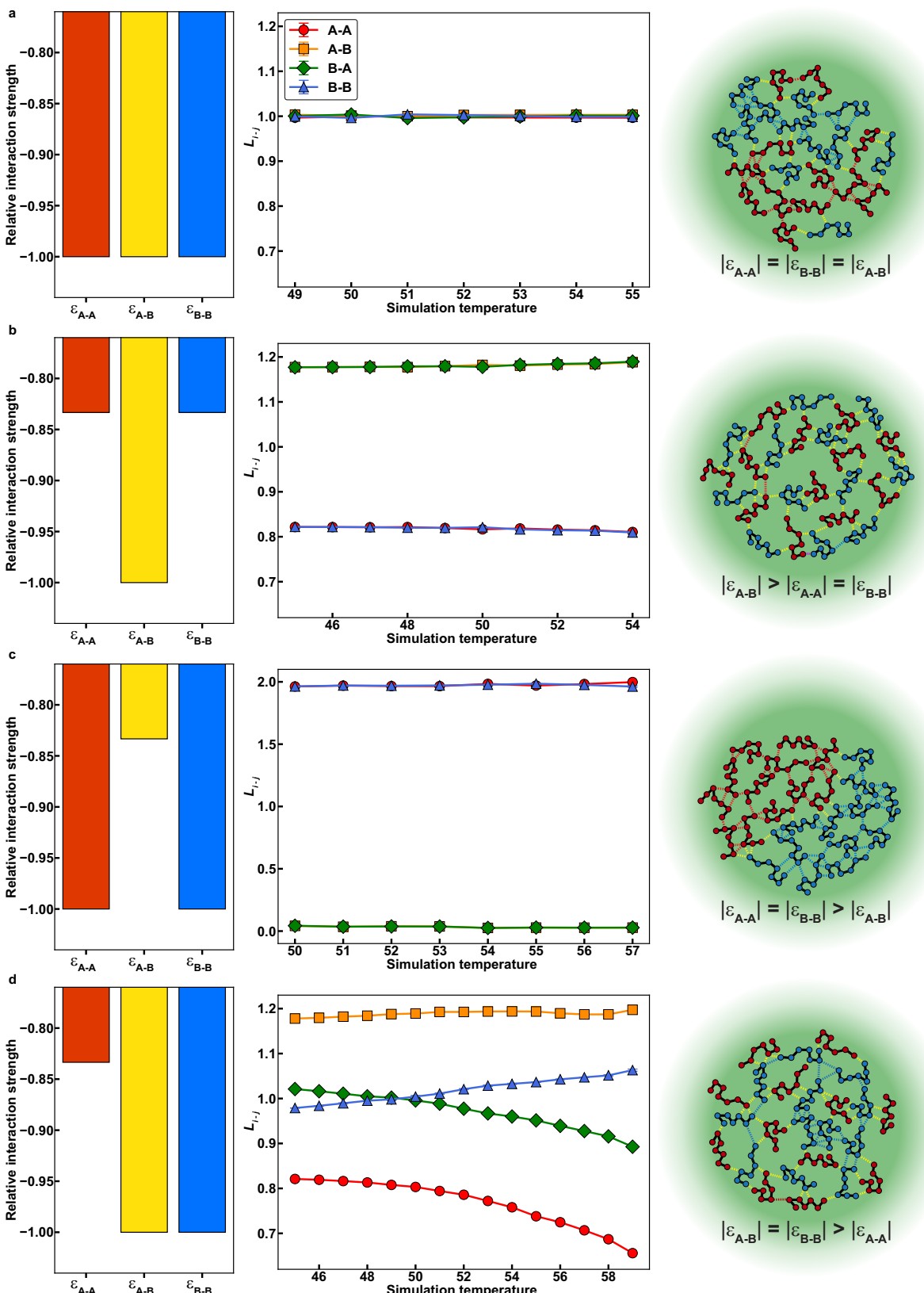

contribute to the internal organization of molecules within condensates[96]. Clearly, physicochemical control of cellular biomolecular condensates, which typically contain dozens of distinct macromolecular species, will involve a complex and dynamic interplay of different types of interactions between macromolecules. Our findings are conceptually reminiscent of the results of Espinosa et al., who used

ultra-coarse-grained patchy colloid models for simulations of complex mixtures of associative macromolecules[97].

If phase separation gives rise to precisely two coexisting phases, then a 1-simplex[39] or tie line will connect the two points that define the coexisting phases. Each phase is defined by concentrations of macromolecules and solution components that enable equalization of

**Fig. 8 | Internal condensate organizations of polymer mixtures depend on the interplay of homotypic vs. heterotypic interactions.** **a–d** Relative interaction energies (left column), values of $L_{i \cdot j}$ (see text) as a function of simulation temperature (central column), and a schematic depicting the predicted condensate organization (right column) for 1:1 mixtures of polymers A and B with various interaction matrices. **a** Homotypic and heterotypic interactions are all equivalent and set to −3.3 in LaSSI. **b** Heterotypic interactions are stronger than homotypic interactions. Heterotypic interactions are set to −3.6 and homotypic interactions to −3.0. **c** Homotypic interactions are stronger than heterotypic interactions. Homotypic interactions are set to −3.6 and heterotypic interactions to −3.0. **d** Homotypic A-A interactions are weaker than homotypic B-B interactions and heterotypic A-B interactions, which are equivalent. Homotypic A-A interactions are set to −3.0 and all other interactions are set to −3.6. $n = 5$ independent simulations with random starting seeds. Error bars for the values of $L_{i \cdot j}$ are standard errors about the mean, though they are typically smaller than the marker size. Source data are provided as a Source Data file.

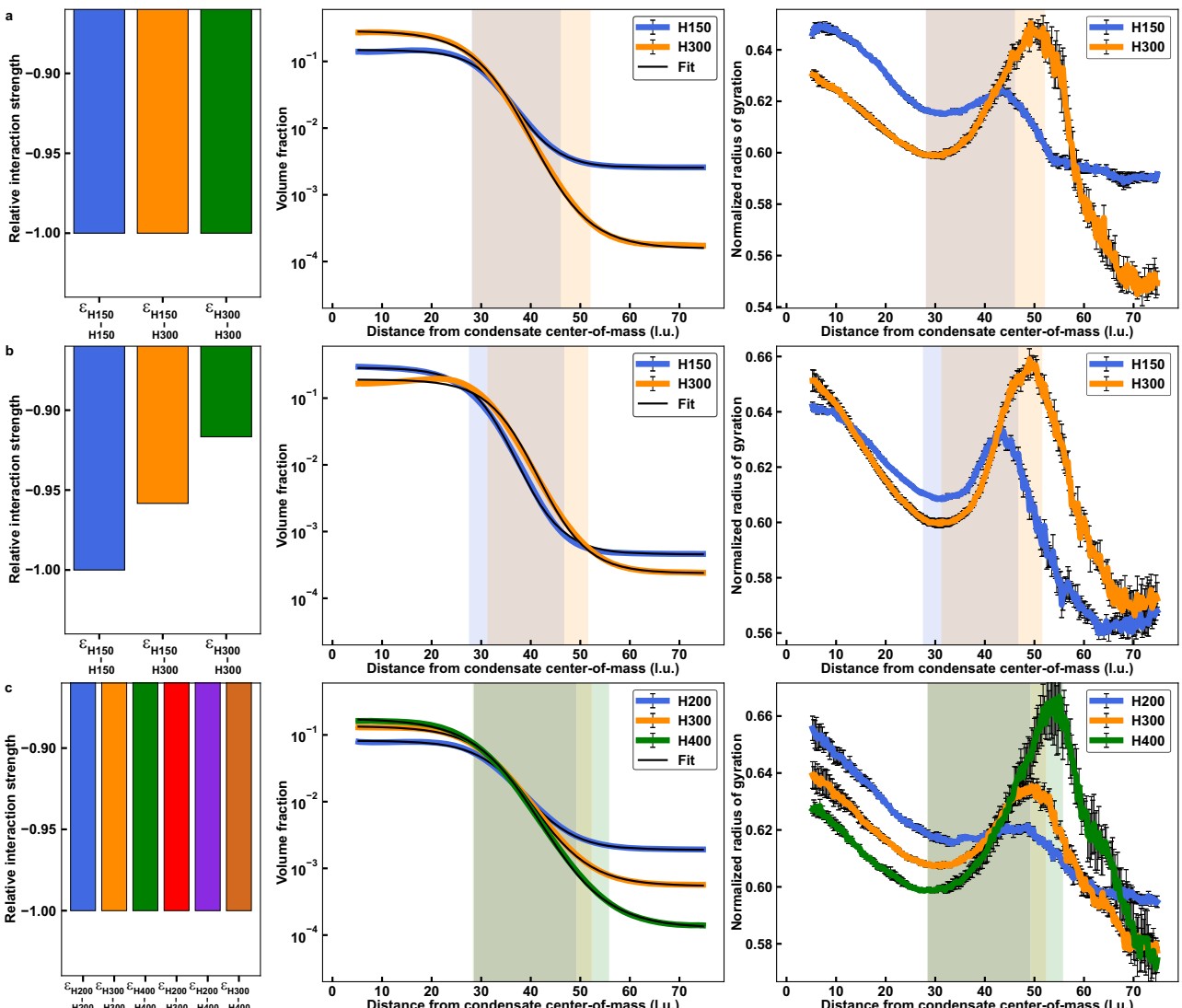

**Fig. 9 | Interfacial features of multi-component condensates depend on polymer lengths and relative interaction energies.** **a–c** Relative interaction energies (left column), radial density plots and associated logistic fits (central column), and normalized radius of gyration plotted against the distance from the condensate center-of-mass (right column) for equal volume fraction mixtures of two or three homopolymers of various lengths and interaction matrices. **a** Mixture of a 150-monomer polymer and a 300-monomer polymer where homotypic and heterotypic interactions are equivalent and set to −3.3 in LaSSI at a simulation temperature of 55. **b** Mixture of a 150-monomer polymer and a 300-monomer polymer where relative interaction strengths are chosen such that the phase diagrams of the two polymers overlay at a simulation temperature of 56. H150-H150 interactions were set to −3.6, H150-H300 interactions to -3.45, and H300-H300 interactions to −3.3 **c** Mixture of a 200-monomer polymer, a 300-monomer polymer, and a 400-monomer polymer where homotypic and heterotypic interactions are equivalent and set to −3.3 at a simulation temperature of 57. $n = 5$ independent simulations with random starting seeds. Error bars are standard errors about the mean. Source data are provided as a Source Data file.

osmotic pressures and chemical potentials across the phases. The slopes of tie lines tell us about the relative contributions of homotypic and heterotypic interactions to phase separation for specific stoichiometric ratios of the macromolecules that drive phase behavior. Using a tie-line analysis, we find that FUS-LCD and A1-LCD engage in strong heterotypic interactions when present in equal mass concentration ratios. However, when present in unequal ratios, homotypic interactions involving the predominant species become the primary driving force for phase separation. This behavior changes for FUS-LCD and A1-LCD+12D mixtures. For this system, phase separation in

mixtures of equal mass concentration ratios is driven mainly by homotypic interactions among FUS-LCD molecules. Tilting the balance of interactions toward heterotypic interactions can be achieved by increasing the ratio of A1-LCD+12D to FUS-LCD. These results highlight the central importance of expression levels and stoichiometric ratios in mixtures of macromolecules[98].

We find that molecules within condensates are organized to maximize the most favorable interactions, be they homotypic or heterotypic. Accordingly, segregative transitions i.e., phase separation drives the formation of condensates, and within condensates associative interactions, i.e., physical crosslinking among favorably interacting molecules, are maximized[39]. Further, molecules that engage in stronger interactions are more likely to be in the core of the condensate as opposed to in the interface. These findings highlight the crucial importance of the coupling of segregative and associative transitions as joint drivers of condensate formation and determinants of condensate internal structures[39].

In agreement with prior work[55], we find that polymers at the condensate interface are likely to adopt expanded conformations. We probed the effects of polymer lengths and found that the degree of expansion increases with polymer length. The strengths of homotypic versus heterotypic interactions and polymer length contribute to preferential localization of molecules at the interface, the degree of chain expansion at the interface, and the thickness of the interface. All polymers show an increase in their $R_g$ values near the condensate interface, and the precise location of the maximal expansion increase is controlled by the species-specific interface. In general, interfaces defined by longer polymers are thicker than those formed by shorter polymers. Interfacial thickness can be altered by modulating the strengths of homotypic and heterotypic interactions, allowing for finer control over spatial localization of each polymer species. The physical properties of interfaces[99] are likely to contribute to capillary forces of condensates[100], interactions between condensates and emulsifiers in vivo[101] or in vitro[102], and chemical reactions that are likely controlled at interfaces[103,104].

Overall, our findings highlight the complex and designable / evolutionarily selected properties[105–107] of biomolecular condensates formed by mixtures of PLCDs. These findings set the foundations for dissecting the phase behaviors and properties of condensates formed by $n$-nary mixtures of multivalent macromolecules. Our work highlights the promise of using simulations for modeling and designing phase behaviors in multicomponent systems[60,93,107–109].

## Methods
### Identifying condensate-associated prion-like low-complexity domains (PLCDs)
To identify the number of proteins with PLCDs that have been located within different cellular condensates, we used DrLLPS[110], a database of condensate-associated proteins. We culled all proteins associated with condensates with at least 40 known components. We then used the algorithm PLAAC[111] to identify proteins in the curated database that contain a PLCD. Within PLAAC, we specified a 50/50 weighting of background probabilities between *H. sapiens* and *S. cerevisiae*. This resulted in 89 distinct condensate-associated proteins with PLCDs. The sequences of these PLCDs were used for analyses that led to the results in Fig. 1a–c.

### Monte Carlo simulations
The coarse-grained model uses one lattice bead per amino acid residue and treats vacant sites as components of the solvent. In this way, solvent is afforded space in the system, but we do not parameterize any explicit interactions between solvent occupied sites and any other sites. Monte Carlo moves are accepted or rejected based on the Metropolis-Hastings criterion such that the probability of accepting a move is the min[1,exp(-$\Delta E/k_BT$)] where $\Delta E$ is the change in total system energy of the attempted move and $k_BT$ is the thermal energy, or simulation temperature. The energetic model used is the same as that described by Farag et al.[55] This model was parameterized using experimental SEC-SAXS data for the conformations of single molecules in dilute phases. The parameters were shown to recapitulate the phase behaviors of over thirty variants of A1-LCD. The model accounts for a variety of interactions, including pi-pi interactions among tyrosine and phenylalanine residues, cation-pi interactions between arginine and aromatic residues, electrostatic interactions, and contributions to the solvation volume from spacer residues. Here, because we perform simulations of multi-component systems, we modified the mean-field electrostatic model, which was originally based on a single-component system. Instead of using a single NCPR value to determine the effect of electrostatics on a pairwise bead-bead interaction, we use the average NCPR of the chains to which the beads belong. By treating electrostatic interactions using this mean-field model, we avoid the risk of misrepresenting electrostatic interactions due to charged residues, salt ions, and solvent, but are unable to capture the effect of charge patterning on the overall phase behavior. The full set of Monte Carlo moves used in each type of simulation performed in this work are shown in Supplementary Table 1. All the move sets and frequencies of moves are as described previously[55].

We performed multi-chain LaSSI simulations at various temperatures. The simulation temperatures are referenced in terms of units where we set $k_B = 1$. Simulations involving FUS-LCD and A1-LCD used a $120 \times 120 \times 120$ cubic lattice with periodic boundary conditions. Those involving homopolymers used a $150 \times 150 \times 150$ cubic lattice with periodic boundary conditions. To speed up condensate formation, the simulations were initialized in a smaller $35 \times 35 \times 35$ cubic lattice, which allows for significantly faster equilibration processes. The number of chains in each simulation was chosen to keep the total volume fraction of beads as close as possible to 0.016. Details of the sequences used in this study are shown in Supplementary Table 2.

A total of $3 \times 10^{10}$ MC steps was deployed for multi-chain simulations at each of the simulation temperatures. Simulations typically equilibrated after about $2 \times 10^9$ steps, as determined by a plateauing of the total system energy. To be conservative, all simulation results were analyzed after the halfway point of $1.5 \times 10^{10}$ steps. Multi-chain simulations were performed with five to ten replicates, with each simulation initiated by a distinct random seed.

### Simulation analyses
The main text and previous work[1] provide a summary of how the LaSSI simulations[2] were setup. We identified the presence of a phase boundary and estimated dilute phase concentrations, dense phase concentrations, interface midpoints, and interface widths using a logistic fit of the radial density[1]. To calculate concentrations, we used the exact prior for a cubic lattice with a given size. To determine chain expansion through the interface, we used radial shells with thickness of 0.25 of a lattice unit. When calculating the radial bins for the chains, rather than using the center-of-mass of a chain and counting each chain one time, we independently counted each bead in the chain using the radial bin of the bead and the radius of gyration of the corresponding chain. This accounts for the fact that a single chain can span multiple bins. The contribution of each bin is weighted by the number of beads of a chain that belong to the bin.

**Crosslinking analysis.** We introduced a crosslinking parameter, $L_{i-j}$ that quantifies the relative likelihood that a molecule from species $i$ interacts with a molecule from species $j$. To calculate $L_{i-j}$, we first calculate $f_{i-j}$, the total number of intermolecular contacts between species $i$ and $j$ within the condensate ($N_{i-j}$), divided by the total number of intermolecular contacts between species $i$ and all

other species:

$$f_{i-j} = \frac{N_{i-j}}{\sum_k N_{i-k}}.$$ (1)

Next, we calculate $f_{j|i}$, the total number of proteins of species $j$ in the condensate from the perspective of a molecule of species $i$ ($N_{j|i}$), divided by the total number of molecules in the condensate from the perspective of a molecule of species $i$:

$$f_{j|i} = \frac{N_{j|i}}{\sum_k N_{k|i}} = \begin{cases} \frac{N_j}{\sum_k N_k - 1} & \text{if } i \neq j \\ \frac{N_j - 1}{\sum_k N_k - 1} & \text{if } i = j \end{cases}.$$ (2)

We subtract 1 from the calculation if $i = j$ to account for the fact that we are only interested in intermolecular interactions. Accordingly, we ignore the protein from whose perspective we are calculating the given value. Finally, we take the ratio of these two values to calculate $L_{i-j}$:

$$L_{i-j} = \frac{f_{i-j}}{f_{j|i}}.$$ (3)

Values of $L_{i-j}$ close to 1 indicate that species $i$ interacts with species $j$ in a manner that is proportional to the number of $i$ and $j$ proteins in the condensate *i.e.*, the proteins are randomly mixed. In contrast, values greater than 1 indicate that species $i$ is more likely to interact with species $j$ than would be expected from random mixing, and values less than 1 indicate that species $i$ is less likely to interact with species $j$ than would be expected.

### In vitro measurements

We used the WT low-complexity domain (LCD) (residues 186-320) of human hnRNPA1 (UniProt: P09651; Isoform A1-A), in which the M9 nuclear localization signal had been mutated, substituting the PY motif with GS (referred to as A1-LCD); we also used a variant in which we further substituted several glycine and serine residues with aspartate residues increasing the aspartate content by 12 residues (A1-LCD+12D); and third, we used the WT LCD (residues 1-214) of FUS (UniProt: P35637). The gene sequences were synthesized as previously described[54], including a gene sequence coding for an N-terminal TEV cleavage site followed by the protein-coding sequence of interest. The protein sequences are shown in Supplementary Table 2. The three proteins were expressed and purified as previously described[53,54,70], and the purified proteins were stored in 6 M GdmHCl (pH 5.5), 20 mM MES at 4 °C until they were buffer exchanged into phase separation buffer.

**Phase separation assays in vitro.** Samples were prepared as described in the Materials and Methods section of the main text and in the recent work of Bremer et al.[3]. The buffer exchange was achieved in two-steps. The total protein concentration was kept at 2.1 mg/mL across all combinations of mixtures to be consistent with the ratios used in the simulations. As an example, for the case when 50% of the protein mixture comprises FUS-LCD, enough A1-LCD was added to bring its final concentration to 1.05 mg/mL, and enough FUS-LCD was added to bring its final concentration to 1.05 mg/mL to ensure a total protein concentration of 2.1 mg/mL. At higher temperatures, the saturation concentrations of FUS-LCD, A1-LCD, and A1-LCD + 12D increase. Accordingly, the input concentrations for the phase separation assays of solutions containing a single protein were also increased to be at least 1.5 times that of the saturation concentration. For example, at 20 °C and 24 °C, the total amount of FUS-LCD used to drive phase separation in the absence of A1-LCD was 7.5 mg / mL and 9.9 mg / mL. Likewise, in the absence of FUS-LCD the total amount of A1-LCD used

to drive phase separation at 20 °C and 24 °C was 5.7 mg/mL 3.1 mg/ml, respectively.

The protein solutions were mixed in 20 mM HEPES (pH 7.0), and 3 M NaCl in 20 mM HEPES (pH 7.0) was spiked into the solution to bring the final NaCl concentration to 150 mM. The samples were incubated at the desired temperatures for 20 min, then centrifuged at this temperature for 5 min at 12,000 rpm to separate the solution into dilute and dense phases. The dilute phase was then transferred to HPLC vials for analysis. A sample of the dense phase was taken for one temperature and diluted 1000-fold with 6 M GdmHCl (pH 5.5), 20 mM MES. The vials of the separated phases were then analyzed by analytical HPLC to assay the component protein concentrations in each phase[4].

**Quantification of coexistence concentrations of mixtures of A1-LCD and FUS-LCD using analytical HPLC.** We determined the dilute and dense phase concentrations ($c_{dilute}$ and $c_{dense}$, respectively) for all A1-LCD and FUS-LCD mixtures using analytical HPLC[4]. Before analyzing the mixtures on the HPLC we ensured that A1-LCD and FUS-LCD elute separately and with sufficient resolution for peak integration. To determine the dilute and dense phase concentrations of each component, we measured a standard curve for each component. The standard curve for each protein was fitted to Eq. 1 in Bremer et al.[4], which was then used to determine $c_{dilute}$ and $c_{dense}$ for each component. At least three replicates per condition were measured.

**Fluorescent labeling of protein samples.** To test for localization of protein species within condensates, each variant was fluorescently labeled at the N-terminus. WT A1-LCD was labeled with Alexa Flour488 (NHS Ester; ThermoFisher), A1-LCD+12D with LD-555 (NHS Ester; Lumidyne Technologies), and FUS-LCD was labeled with LD-655 (NHS Ester; Lumidyne Technologies). The protein samples were labeled under denaturing conditions, in 50 mM Phosphate buffer (pH 7.0), 6M GdmHCl, and quenched with 20mM Tris.

**Confocal fluorescence microscopy.** The extent of co-localization of A1-LCD and FUS-LCD for the different A1-LCD and FUS-LCD mixtures was examined using a Zeiss LSM 980 Airyscan 2. Samples were prepared as described in the phase separation assay, but each protein was doped with a fluorescently labeled fraction of itself. 2 µL of the mixture of interest was placed between two coverslips sandwiched with 3 M 300 LSE high-temperature double-sided tape (0.34 mm) with a window for microscopy cut out. All measurements were performed in 20 mM HEPES (pH 7.0), 150 mM NaCl at room temperature, and image analysis was performed using Fiji (Version 2.1.0/1.53o)[5].

### Reporting summary

Further information on research design is available in the Nature Portfolio Reporting Summary linked to this article.

## Data availability

All experimental data are available at the following Pappu Lab GitHub repository: https://github.com/Pappulab/Data-and-Analysis-for-PLCD-Mixtures. Details of the sequences used for analysis that yielded results in Fig. 1 are available via Github (URL referenced in the previous sentence). Protein expression constructs for in vitro measurements are available from Addgene. Source data are provided with this paper.

## Code availability

All code used to analyze simulations are available at the following Pappu Lab GitHub repository: https://github.com/Pappulab/Data-and-Analysis-for-PLCD-Mixtures. Farag, M., Borcherds, W., Bremer, A., Mittag, T., & Pappu R. (2023). Phase Separation of Protein Mixtures is Driven by the Interplay of Homotypic and Heterotypic Interactions. Code available at https://doi.org/10.5281/zenodo.8242385.

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

## Acknowledgements

This work was supported by grants from the US National Institutes of Health (RO1NS121114 to RVP and TM), the St. Jude Research Collaborative on the Biology and Biophysics of RNP granules (to RVP and TM), and the Air Force Office of Scientific Research (FA9550-20-1-0241 to RVP). We are grateful to Daoyuan Qian and Tuomas Knowles for stimulating discussions.

## Author contributions

M.F. and R.V.P. came up with the project idea, designed the simulations, and worked together on all the analyses for computations. M.F. developed the system-specific and generic computational models and performed all the simulations. T.M., W.M.B., and A.B. designed the experiments. W.M.B. and A.B. provided the experimental data. M.F., W.M.B., A.B., T.M., and R.V.P. analyzed the data. M.F. and R.V.P. wrote the manuscript and all authors edited the manuscript.

## Competing interests

R.V.P. is a member of the Scientific Advisory Board and a shareholder of Dewpoint Therapeutics. The work reported here was not influenced by this affiliation. The remaining authors declare no competing interests.
