## [Peer Review File · Nature Communications]

REVIEWER COMMENTS

Reviewer #1 (Remarks to the Author):

The manuscript by Farag et al. examines the interplay of homotypic and heterotypic interactions of low-complexity domains in determining phase separation in a two-component system, using FUS-LCD and A1-LCD as a model system. They demonstrate that complementary electrostatic interactions between the two proteins enhance their phase separation, providing evidence that heterotypic interactions play critical roles in determining phase boundary shape. Additionally, they explore how stoichiometry and molecular sequence impact phase separation, using both computational simulations and in vitro experiments. The manuscript also investigates the inner structure of condensates as a function of homotypic-heterotypic interactions.

Overall, the manuscript is insightful, convincing, and comprehensive. It addresses fundamental yet crucial topics in the phase separation field, which deepens our understanding of condensate formation. Undoubtedly, the findings in this manuscript could generalize to other multi-component systems, thus is highly valuable and impactful. I have a few comments that could hopefully help improve the manuscript, but it is already in good shape.

Major Points:

1. It seems to me that the shape of the phase boundary in Fig. 2C is determined by whether component A and B is attractive or repulsive, not by the interplay between homotypic and heterotypic interactions. Based on the LaSSI method in the main text line 524 “we use the average NCPR of the chains to which the beads belong”, there should be no homotypic interactions at all. In another word, regardless of how strong homotypic interaction is, if there’s any attractive heterotypic interaction between FUS-LCD and A1-LCD, the phase boundary will be concave. If repulsive, then convex.
2. To show that heterotypic interaction dominates in the phase separation of FUS-LCD and A1-LCD, maybe it’s helpful to add the saturation concentration of each molecule in Fig. 3 as a control. If the addition of the other molecule significantly reduces the phase separation threshold, then it is crystal clear that heterotypic interaction dominates.
3. It might be helpful to add another FUS-LCD variant construct with neutral or positive charges in simulation and/or in vitro experiments. This, together with A1+12D experiment, will provide orthogonal evidence showing that the electrostatic interaction between FUS-LCD and A1-LCD are the main heterotypic interactions driving phase separation.
4. The interpretation of Figure. 3D does not make sense to me. It looks like it’s very close to the additive line, instead of convex. The reason behind it is complex: In real in vitro system, there are homotypic interaction of both FUS and A1, there are self-, homo- and hetero- electrostatic interactions, there are

aromatic interactions. All these interactions combined, the heterotypic interaction between FUS and A1 is almost net zero.

5. Comparing Fig. 3c-d and Fig. 2 d-e, it seems that the phase boundaries of experimental assays are more concave. My guess is that it's mainly the aromatic residue interactions that offset the repulsion between charged residues, resulting in a straight-line phase boundary.

6. The tie line part seems interesting. I wonder if the authors would consider an in vitro experiment having both FUS-LCD and A1-LCD and measure the slope? I think ratio of 1:1, 1:3 and 3:1 should be good enough, just as a proof of concept.

7. Frankly I feel that the Fig. 7-9 should be published in another manuscript or wrapped in the supplementary, because this will make the main message sound and clear: heterotypic and homotypic interactions together shape the phase behavior of multi-component systems. I am personally fascinated by the results of how the internal organization of condensates are determined by relative strengths of homotypic and heterotypic interactions. It will be a more complete story with experimental data. With that being said, I am also fine with what it is now.

Minor points:

1. The resolution of the main figures can be improved

2. The total concentration was kept at 2.1 mg/ml. If this is the case, then changing the relative proportions of two proteins is going to change the total molar concentrations, because the molecular weights of two proteins are different

Reviewer #2 (Remarks to the Author):

The manuscript "Phase Separation in Mixtures of Prion-Like Low Complexity Domains is Driven by the Interplay of Homotypic and Heterotypic Interactions" by Farag et al. investigates the effect of composition and stoichiometry on the stability of condensates made of mixtures of the hnRNPA1-LCD and the FUS-LCD via a combination of simulations and experiments. I very much enjoyed reading this article. The simulations and experimental results are very exciting as they highlight the high dependency of the phase behaviour of proteins on the environment. The work demonstrates how the phase behaviour of proteins within multi-component environments is more complex than what can be anticipated by simply looking at the phase behaviour of single-component systems. I would be supportive of the publication of this work in Nature Communications. However, I have a few suggestions and comments that I would like the authors to address first.

1. The simulations reveal that the 50:50 and 25:75 mixtures of hnRNPA1-LCD and FUS-LCD have lower dilute phase coexistence densities at a range of temperatures than those exhibited by either of the components in pure form. Such lower densities typically imply higher critical solution temperatures and larger coexistence regions, and hence, are indicative of a larger range of stability of the condensed phases. The authors explain that the observed favoured phase separation for the mixtures results from additional associative electrostatic interactions between the two proteins. The predictions of the model agree perfectly with the experimental results. However, the details on how exactly the electrostatic interactions are modelled, and how the model achieves the correct balance of electrostatic and non-electrostatic interactions to recapitulate the experimental observations so perfectly are very brief. Thus, it is not completely convincing to me whether the simulations have independently predicted the experimental observations, or whether the model has been designed to achieve such an excellent agreement with the experiments a priori. For instance, it seems that in the model a sufficiently negative energy term, defined to be contributed by electrostatics, was added to account for additional heterotypic interactions between hnRNPA1-LCD and FUS-LCD, so that the mixture would naturally exhibit enhanced phase separation. How is competition between the two proteins for homotypic interaction sites considered? The lack of details on the model parametrisation for the hnRNPA1-LCD and FUS-LCD mixture also makes it hard to discern if the experimental results are unequivocally accounted for by additional electrostatic interactions among the two protein components or if other forces are also at play. The experiments show that when salt is increased from 150 to 300 mM NaCl, the lower coexistence densities of the 1:1 mixture increase, and the saturation concentrations of the pure systems decrease; such results are indeed indicative that heterotypic electrostatics dominate the enhanced phase separation of the mixture. If it wasn't for these salt-dependent experimental observations, how were alternative scenarios ruled out from the model, e.g. adding an additional term for stronger hnRNPA1-LCD and FUS-LCD non-electrostatic heterotypic interactions instead of or on top of the additional electrostatic term?

2. I found the last section on the trends emerging from simpler polymers very interesting. Could the authors explain how the interplay between mixture stoichiometry and variance in homotypic/heterotypic interactions affects the internal organisation of the two-component condensates? Given the rich mechanical properties expected to be exhibited by IDPs (e.g. various degrees of flexibility, flickering secondary structural motifs), besides the variance in homotypic and heterotypic interaction strengths and polymer length, would the authors be able to assess how the mechanical properties of the polymers affect the internal organisation of the two-component systems? The formation of beta-sheets among FUS-LCDs inside condensates has been suggested to drive single-component FUS condensates to form hollow architectures, so I wonder if varying the mechanical potential of the model polymers might give rise to other unexpected and rich condensate behaviours.

Minor comments:

*Would it be possible to add an estimate of the critical parameters for each case in the simulation binodals?

*There are no units on the simulation plots. Can these be added or if not, can the authors explain why units are not needed/used?

Reviewer #3 (Remarks to the Author):

In this manuscript Farag and co-workers investigate the phase separation properties of solutions of two different prion-like low-complexity domains. They analyze how their compositions influence phase separation propensity as well as the organization of the condensate. For this the authors used coarse-grained molecular simulations, in some cases validated experimentally, obtaining insights into the key role played by the relative strengths of homo and heterotypic interactions.

The topic chosen by the authors is relevant, the methodology is mostly appropriate, the results are correctly interpreted, and the findings will be of general interest to scientists studying intrinsically disordered proteins and biomolecular phase transitions. Before the work is published in Nature Communications, however, it would be necessary for the authors to address the following points, including some related to the presentation of the results.

Main points

- 1 - The authors state that, in the absence of heterotypic interactions, the C_{sat} (in units of total mass concentration) of the relevant solutions is a linear combination of those of the pure components (Fig. 2C). Although intuitive, this statement should be substantiated, either from theory or, even better, by carrying out a molecular simulation where the energy term corresponding to heterotypic interactions is removed.
- 2 - The deviation from this expectation for wild type A1-LCD is large both in the simulations (Fig. 2C) and the experiments performed in vitro (Fig. 3C) but very small for the mutant A1-LCD+12D. Why is the deviation from expectation for A1-LCD+12D so small, especially experimentally? The readers will greatly appreciate that the authors propose an explanation or, even better, that they address this question with a computational experiment.
- 3 - The results obtained for A1-LCD+12D presented in Fig. 6 are those obtained experimentally. It would however be interesting to see the simulation results as well, as in Fig. 5; this will give the readers additional confidence that the simulations provide realistic representations of the condensates.

4 - The results presented in Fig. 8 are interesting and help understand those presented in Fig. 7C. The scenario represented in panel C of Fig. 8, where heterotypic interactions are weak, seems to correspond to the formation of two co-existing immiscible liquid phases. According to the authors this scenario should lead to a phase diagram where C_{sat} , C_{tot} and C_{dense} are not in a straight line. The authors should clarify this by computing the phase diagram of this system.

Minor points

1 - It would be very helpful for the readers if the authors computed the full phase diagram for the A1-LCD FUS-LCD system from simulations. This would not generate new information but would make the work easier to follow for non-specialized readers of a journal such as Nature Communications.

2 - Contact maps of the simulations of 0, 50 and 100% FUS-LCD mixtures could be shown to strengthen the conclusion that heterotypic electrostatic interactions enhance phase separation.

3 - The ionic strength experiment in Fig. 3F is very nice and informative. However, the differences in the C_{dilute} of a 50% mixture are small. Maybe having a third condition (NaCl concentration higher than 300 mM) to see if the magnitude of the change in C_{dilute} increases further would help.

4 - The manuscript is well-written but the figures need some work. Some of them could be moved to the Supplementary Material, some others could be merged and, in some cases, they could be a little more clear: in any case 9 Figures appears excessive for Nature Communications.

Responses to reviewers

Responses to comments of Reviewer 1

Comment 1: *The manuscript by Farag et al. examines the interplay of homotypic and heterotypic interactions of low-complexity domains in determining phase separation in a two-component system, using FUS-LCD and A1-LCD as a model system. They demonstrate that complementary electrostatic interactions between the two proteins enhance their phase separation, providing evidence that heterotypic interactions play critical roles in determining phase boundary shape. Additionally, they explore how stoichiometry and molecular sequence impact phase separation, using both computational simulations and in vitro experiments. The manuscript also investigates the inner structure of condensates as a function of homotypic-heterotypic interactions.*

Overall, the manuscript is insightful, convincing, and comprehensive. It addresses fundamental yet crucial topics in the phase separation field, which deepens our understanding of condensate formation. Undoubtedly, the findings in this manuscript could generalize to other multi-component systems, thus is highly valuable and impactful. I have a few comments that could hopefully help improve the manuscript, but it is already in good shape.

Response to comment 1: We thank the reviewer for their thoughtful summary and positive comments on the manuscript.

Comment 2: *It seems to me that the shape of the phase boundary in Fig. 2C is determined by whether component A and B is attractive or repulsive, not by the interplay between homotypic and heterotypic interactions. Based on the LaSSI method in the main text line 524 “we use the average NCPR of the chains to which the beads belong”, there should be no homotypic interactions at all. In another word, regardless of how strong homotypic interaction is, if there’s any attractive heterotypic interaction between FUS-LCD and A1-LCD, the phase boundary will be concave. If repulsive, then convex.*

Response to comment 2: We acknowledge the reviewer’s comment that our wording may be imprecise, though we also note that there may be some clarification needed to make sure that we are on the same page as the reviewer. Importantly, the main drivers of homotypic and heterotypic interactions are interactions between aromatic stickers. The electrostatic interactions play a secondary role in the overall interaction hierarchy. We use the same model as previously described (<https://doi.org/10.1038/s41467-022-35370-7>), which accounts for a variety of interactions, not just electrostatic interactions. Thus, the concavity of the phase diagram derives from the totality of heterotypic interactions (including aromatic-aromatic, aromatic-arginine, electrostatic interactions, and differential contributions of spacer residues including glycine, serine, threonine, asparagine, and glutamine) being more dominant than the combinations of homotypic interactions. Conversely, the convexity derives from the heterotypic interactions being weaker than the homotypic interactions. The heterotypic interactions do not necessarily need to be repulsive for the phase diagram to be convex. In fact, if they were repulsive, one would likely see two distinct condensates as opposed to one co-localized condensate. Please note that we have previously shown that molecules with highly positive or negative net charges have less attractive / more repulsive homotypic electrostatic interactions than those with near-neutral electrostatic interactions (<https://doi.org/10.1038/s41557-021-00840-w>, <https://doi.org/10.1038/s41467-022-35370-7>) (). Taking these points together, the comment “there should be no homotypic interactions at all” is not valid. To clarify these points, we have included more information on the model in the Methods.

Comment 3: *To show that heterotypic interaction dominates in the phase separation of FUS-LCD and A1-LCD, maybe it's helpful to add the saturation concentration of each molecule in Fig. 3 as a control. If the addition of the other molecule significantly reduces the phase separation threshold, then it is crystal clear that heterotypic interaction dominates.*

Response to comment 3: In Figure 3, we show the individual saturation concentrations of FUS-LCD, A1-LCD, and A1-LCD +12D throughout panels B, C, D, and F. For example, in panel F, we show the dilute phase concentration in the presence of 0% FUS-LCD and 100% FUS-LCD. These values correspond to the saturation concentrations of A1-LCD and FUS-LCD, respectively. In the same figure, we show that a 50/50 mixture of FUS-LCD and A1-LCD has a lower total dilute phase concentration than either protein on its own. These data, which were always present, fully address the reviewer's comment.

Comment 4: *It might be helpful to add another FUS-LCD variant construct with neutral or positive charges in simulation and/or in vitro experiments. This, together with A1+12D experiment, will provide orthogonal evidence showing that the electrostatic interaction between FUS-LCD and A1-LCD are the main heterotypic interactions driving phase separation.*

Response to comment 4: We have now performed a simulation of a positively charged variant of FUS-LCD and included our results in the SI. These results agree with our prior results showing that the homotypic interactions are more favorable than the heterotypic interactions if there is a generic, mean-field-like electrostatic repulsion between molecules A and B.

Comment 5: *The interpretation of Figure. 3D does not make sense to me. It looks like it's very close to the additive line, instead of convex. The reason behind it is complex: In real in vitro system, there are homotypic interaction of both FUS and A1, there are self-, homo- and hetero-electrostatic interactions, there are aromatic interactions. All these interactions combined, the heterotypic interaction between FUS and A1 is almost net zero.*

Response to comment 5: We point the reviewer to our response to comment 2 for an in-depth analysis of this panel. As mentioned by the reviewer, there is a hierarchy of interactions. The overall finding is that the heterotypic interactions are not net zero, but rather they are on a par with the homotypic interactions of either protein species in the system. In other words, FUS-LCD molecules, and A1-LCD +12D molecules interact with themselves about as much as they interact with each other.

Comment 6: *Comparing Fig. 3c-d and Fig. 2 d-e, it seems that the phase boundaries of experimental assays are more concave. My guess is that it's mainly the aromatic residue interactions that offset the repulsion between charged residues, resulting in a straight-line phase boundary.*

Response to comment 6: As mentioned in our response to comment 2, the LaSSI simulations account for a variety of interactions, not just electrostatic interactions. The apparent disparity between the simulations and experiments likely derives from two aspects of the simulations, which at this juncture must be viewed as weaknesses: First, as we have noted repeatedly, the methodology used to parameterize the electrostatic interactions is crude. It rescales interactions based on the NCPR values and is a purely mean-field approach that transfers the V-shape-based result from Farag et al. (<https://doi.org/10.1038/s41467-022-35370-7>)), to the current setting. Second, this cannot be corrected by the inclusion of site-specific, Debye-Hückel type interactions because results from our unpublished measurements show that there is differential partitioning of solution

ions that complicates the way ion effects are to be handled. Therefore, the difference between the computed and measured concavities are likely to do with the hidden components, namely the non-trivial effects of solution ions. We have clarified this weakness of the model in the Methods and in the revised Discussion section.

Comment 7: *The tie line part seems interesting. I wonder if the authors would consider an in vitro experiment having both FUS-LCD and A1-LCD and measure the slope? I think ratio of 1:1, 1:3 and 3:1 should be good enough, just as a proof of concept.*

Response to comment 7: Perhaps we are misunderstanding the reviewer, but we believe that this is precisely what we show in Figure 5D-F. We welcome further clarification in case we have misunderstood the reviewer's comment.

Comment 8: *Frankly I feel that the Fig. 7-9 should be published in another manuscript or wrapped in the supplementary, because this will make the main message sound and clear: heterotypic and homotypic interactions together shape the phase behavior of multi-component systems. I am personally fascinated by the results of how the internal organization of condensates are determined by relative strengths of homotypic and heterotypic interactions. It will be a more complete story with experimental data. With that being said, I am also fine with what it is now.*

Response to comment 8: We appreciate the reviewer's interest in Figures 7-9. While we understand the reviewer's suggestion, we believe that these data belong in this manuscript. A major message of this manuscript is the importance of heterotypic and homotypic interactions in understanding the complete phase behavior of mixtures of proteins. In these figures, we directly examine how these types of interactions affect the internal organizations of the resulting condensates, which is directly relevant to the overall phase behavior.

Comment 9: *The resolution of the main figures can be improved.*

Response to comment 9: We regret for any problems that were caused with figure resolution in the original submission and have made sure that the figure resolution is in line with the requirements of the journal.

Comment 10: *The total concentration was kept at 2.1 mg/ml. If this is the case, then changing the relative proportions of two proteins is going to change the total molar concentrations, because the molecular weights of two proteins are different.*

Response to comment 10: Yes, this is correct. We have made sure to maintain a constant mass concentration as opposed to a molar concentration to account for the molecular weight differences between different protein species.

Responses to comments of Reviewer 2

Comment 1: *The manuscript "Phase Separation in Mixtures of Prion-Like Low Complexity Domains is Driven by the Interplay of Homotypic and Heterotypic Interactions" by Farag et al. investigates the effect of composition and stoichiometry on the stability of condensates made of mixtures of the hnRNPA1-LCD and the FUS-LCD via a combination of simulations and experiments. I very much enjoyed reading this article. The simulations and experimental results are very exciting as they highlight the high dependency of the phase behaviour of proteins on the environment. The work demonstrates how the phase behaviour of proteins within multi-component environments is more complex than what can be anticipated by simply looking at the phase behaviour of single-component systems. I would be supportive of the publication of this work in*

Nature Communications. However, I have a few suggestions and comments that I would like the authors to address first.

Response to comment 1: We thank the reviewer for their positive comments and assessments of our work.

Comment 2: *The simulations reveal that the 50:50 and 25:75 mixtures of hnRNPA1-LCD and FUS-LCD have lower dilute phase coexistence densities at a range of temperatures than those exhibited by either of the components in pure form. Such lower densities typically imply higher critical solution temperatures and larger coexistence regions, and hence, are indicative of a larger range of stability of the condensed phases. The authors explain that the observed favoured phase separation for the mixtures results from additional associative electrostatic interactions between the two proteins. The predictions of the model agree perfectly with the experimental results. However, the details on how exactly the electrostatic interactions are modelled, and how the model achieves the correct balance of electrostatic and non-electrostatic interactions to recapitulate the experimental observations so perfectly are very brief. Thus, it is not completely convincing to me whether the simulations have independently predicted the experimental observations, or whether the model has been designed to achieve such an excellent agreement with the experiments a priori. For instance, it seems that in the model a sufficiently negative energy term, defined to be contributed by electrostatics, was added to account for additional heterotypic interactions between hnRNPA1-LCD and FUS-LCD, so that the mixture would naturally exhibit enhanced phase separation. How is competition between the two proteins for homotypic interaction sites considered? The lack of details on the model parametrisation for the hnRNPA1-LCD and FUS-LCD mixture also makes it hard to discern if the experimental results are unequivocally accounted for by additional electrostatic interactions among the two protein components or if other forces are also at play. The experiments show that when salt is increased from 150 to 300 mM NaCl, the lower coexistence densities of the 1:1 mixture increase, and the saturation concentrations of the pure systems decrease; such results are indeed indicative that heterotypic electrostatics dominate the enhanced phase separation of the mixture. If it wasn't for these salt-dependent experimental observations, how were alternative scenarios ruled out from the model, e.g. adding an additional term for stronger hnRNPA1-LCD and FUS-LCD non-electrostatic heterotypic interactions instead of or on top of the additional electrostatic term?*

Response to comment 2: Details of how the model was developed were furnished in our earlier work (<https://doi.org/10.1038/s41467-022-35370-7>). Based on the reviewer's comments, we have reproduced all the details in the expanded methods section and in the expanded and revised SI. The model was parameterized and jackknifed using experimental SAXS data and recapitulates the phase behavior of A1-LCD and ~30 distinct variants. This model accounts for aromatic-aromatic interactions, aromatic-arginine interactions, electrostatic interactions, and differential contributions of spacer residues including glycine, serine, threonine, asparagine and glutamine. For simplicity, and to avoid the possibility of misrepresenting the intricate details of electrostatic interactions, we chose to treat electrostatic interactions in a mean-field manner that only considers the net charge per residue of a given molecule. In that work, we only applied this model to single-component systems, so all molecules had the same net charge per residue. For the current work, because we are specifically interested in multi-component systems, we adapted this mean-field model to use the *average* net charge per residue of the interacting molecules. Otherwise, the model used in this work is the same as that used in our prior work. **Notably, all the simulations in this work were performed prior to the experiments, so no aspect of the computational model was**

designed to match experimental data. Rather, the experiments were performed specifically to validate the computational results. We hope that these clarifications assuage the reviewer's concerns. Importantly, the imperfections of the model (note that all models are imperfect) become clear in the comparisons of the computed and measured phase boundaries. The low concentration arm of the phase boundary is considerably more concave in the experiments than in the simulations.

Comment 3: *I found the last section on the trends emerging from simpler polymers very interesting. Could the authors explain how the interplay between mixture stoichiometry and variance in homotypic/heterotypic interactions affects the internal organisation of the two-component condensates? Given the rich mechanical properties expected to be exhibited by IDPs (e.g. various degrees of flexibility, flickering secondary structural motifs), besides the variance in homotypic and heterotypic interaction strengths and polymer length, would the authors be able to assess how the mechanical properties of the polymers affect the internal organisation of the two-component systems? The formation of beta-sheets among FUS-LCDs inside condensates has been suggested to drive single-component FUS condensates to form hollow architectures, so I wonder if varying the mechanical potential of the model polymers might give rise to other unexpected and rich condensate behaviours.*

Response to comment 3: In response to the first half of this comment, we have included more data investigating how mixture stoichiometry affects the internal condensate organization for the two-component systems in Figure 8. These data are shown in the SI. Regarding the second half of the comment, we respectfully submit that examining the mechanical properties of polymers is beyond the scope of the current manuscript. In recent work, we have shown that the network structures extracted from our simulations are sufficient for reproducing experimentally measured, frequency dependent dynamical moduli for nascent condensates formed by A1-LCD and several sequence variants of this system (Alshareedah, Borchers, Cohen et al., bioRxiv 2023, <https://www.biorxiv.org/content/10.1101/2023.04.06.535902v1>). However, the viscoelastic properties of condensates that show physical aging require a whole new framework. As noted in our recent preprint, these systems undergo aging when the spacers are mutated to weaken the driving forces for phase separation. And upon aging, the condensates transition from being terminally viscous Maxwell fluids to terminally elastic Kelvin-Voigt solids. Interestingly, this type of aging is not a glass transition, nor does it appear to involve conversions to beta sheets. These findings, preprinted and currently under review, necessitate a whole new way of thinking about aging that goes beyond a beta-sheet-centric view. Further, our coarse-grained model, indeed any single bead per residue model, does not have the resolution required for observing secondary structure transitions without imposing some form of nematic order parameter.

The issue of chain flexibility / rigidity is an interesting one. This is an entire investigation unto itself and beyond the scope of the current manuscript. Please note that this is something of a generic question and less specific to IDPs that are bereft of proline residues. After all, the intrinsic flexibility of polypeptides is altered either by changing the Gly / Pro content or by incorporating rigid secondary structures or super secondary structures. These type of rod-coil systems are being investigated in the context of a different manuscript that is currently being revised.

Comment 4: *Would it be possible to add an estimate of the critical parameters for each case in the simulation binodals?*

Response to comment 4: The nature of the critical region is, for us at least, an open and unresolved issue. It is common practice to assume that these systems belong to the same universality class as the 3D Ising model. However, the validity of this assumption has never been independently tested. In parallel efforts, we have embarked on a systematic assessment of how the order parameters (note the plurality) scale with distance to the critical point. Our results suggest that the exponents are not those of a 3D Ising model. Further, there is a crossover between the mean-field and critical regime that requires the use of finite size rescaling methods for going around critical points and connecting the mean-field and critical regimes. Pending a complete understanding of these important issues, we think it premature to impose a model and quantify criticality knowing full well that this model is most likely incorrect / incomplete. We recently wrote about these issues in two forums, including our recent publication (<https://doi.org/10.1038/s41467-022-35370-7>) and in a Chemical Reviews contribution. The relevant paragraph from the review is reproduced here for completeness from (<https://doi.org/10.1021/acs.chemrev.2c00814>)

In the (ϕ, T) or (c, T) plane, the critical point is defined in terms of a critical volume fraction ϕ_c and the critical temperature T_c . In a system that undergoes a UCST-type phase transition, the width of the two-phase regime for a given temperature T can be written as: $(\phi_{\text{dense}} - \phi_{\text{sat}}) \propto (T_c - T)^\beta$. Here, β is the relevant critical exponent that describes how the order parameter vanishes as T_c is approached. For phase separation, the order parameter is the $(\phi_{\text{dense}} - \phi_{\text{sat}})$, which is the width of the two-phase regime. The critical exponent is thought to be universal in its applicability to any system undergoing phase separation.

The classical mean-field theory of Landau²³¹ predicts a value of $\beta = 0.5$. In accord with its mean-field nature, the Flory-Huggins theory also yields a value of $\beta = 0.5$ ¹⁶. Simulations of the Ising model show that $\beta \approx 0.3264$ for $d = 3$ ⁷⁴. Many computational studies have assumed that phase separation belongs to the same universality class as an Ising model in $d = 3$ ^{229, 232}. Accordingly, one often sees an imposition of the expectation that the order parameter will scale as $(T_c - T)^{0.33}$. This is then used to extract T_c through numerical fitting²³³ of the computed binodals^{229, 230, 232}. Are these expectations valid? Data for polystyrene in methylcyclohexane²³⁴ show the presence of two regimes, a mean-field regime below T_c , where the order parameter scales as $(T_c - T)^{0.5}$, and a critical regime, where the order parameter scales as $(T_c - T)^{0.33}$. These data also show a clear crossover between the two regimes. The implication is that there are two different regimes for the density fluctuations, one where short-range, Flory-style interactions dominate, and another regime near the critical point, where the density fluctuations are likely to be large enough to be divergent.

Unlike the data for polystyrene in methylcyclohexane²³⁴, none of the measurements reported in the literature for finite-sized, multivalent associative biomacromolecules show this crossover behavior. Instead, the data appear to be well described by the mean-field exponent of 0.5⁶⁴. Therefore, the data to date suggest that the width of the two-phase regime scales as $(T_c - T)^{0.5}$ and not with an Ising-like exponent of 0.33. This would be in accord with the expectation of phase separation being a first-order transition, dominated by short-range interactions that are the defining hallmark of descriptions of phase behaviors in Flory-style theories. Alternatively, if phase separation involves dominant contributions from long-range interactions and large-scale fluctuations, which would be the hallmark of a continuous transition, then the observed exponent should be 0.33. However, this does not appear to be the case. Note that the discussion here applies only to the critical regime for phase separation. As shown in **Figure 13**, there will be two critical

points in a PSCP-type process, namely one corresponding to the terminus of the liquid-liquid phase transition and one corresponding to the terminus of the percolation line. What is currently missing is a rigorous description of how these critical temperatures are determined and how the gap between these values changes based on polymer architectures and interaction strengths.

Please also note that contrary to some recent assertions in the literature, our groups have not succeeded in obtaining reliable estimates of the critical parameters – a failure that highlights the difficulties and challenges posed by the divergent fluctuations at the critical point. We have estimates from fits to Flory-Huggins and other theories, but these are most certainly incorrect descriptions of the critical regime.

Comment 5: *There are no units on the simulation plots. Can these be added or if not, can the authors explain why units are not needed/used?*

Response to comment 5: The simulation concentrations are shown as volume fractions, which means they are, appropriately, unitless. We have clarified this point in the revised manuscript. This is a unitless parameter between 0 and 1. Similarly, the temperatures are in units of simulation temperature, which are described in the Methods. Here, “0” corresponds to absolute zero, like Kelvin, but a single unit of simulation temperature does not correspond to any other temperature unit. In the past, we have used a scaling factor to convert from units of simulation temperature to Kelvin, though we have not done so here to avoid the assumption of transferability and incorrect confluences of computational and experimental data.

Responses to comments of Reviewer 3

Comment 1: *In this manuscript Farag and co-workers investigate the phase separation properties of solutions of two different prion-like low-complexity domains. They analyze how their compositions influence phase separation propensity as well as the organization of the condensate. For this the authors used coarse-grained molecular simulations, in some cases validated experimentally, obtaining insights into the key role played by the relative strengths of homo and heterotypic interactions.*

The topic chosen by the authors is relevant, the methodology is mostly appropriate, the results are correctly interpreted, and the findings will be of general interest to scientists studying intrinsically disordered proteins and biomolecular phase transitions. Before the work is published in Nature Communications, however, it would be necessary for the authors to address the following points, including some related to the presentation of the results.

Response to comment 1: We thank the reviewer for their thoughtful summary and review.

Comment 2: *The authors state that, in the absence of heterotypic interactions, the c_{sat} (in units of total mass concentration) of the relevant solutions is a linear combination of those of the pure components (Fig. 2C). Although intuitive, this statement should be substantiated, either from theory or, even better, by carrying out a molecular simulation where the energy term corresponding to heterotypic interactions is removed.*

Response to comment 2: We note that the c_{sat} of the relevant solution is a linear combination of those of the pure components when the heterotypic interactions are equivalent to the homotypic interactions, not when the heterotypic interactions are absent, as suggested by the reviewer. If the heterotypic interactions were absent, the distinct components would likely not co-phase separate into a single condensate. We have now included an example of the behavior described by the reviewer in the SI. Here, we perform simulations of A1-LCD molecules that are labeled as Type

A or as Type B. In this case, the homotypic and heterotypic interactions are all equivalent among the two types of molecules, resulting in a straight line for the dilute arm of the phase diagram.

Comment 3: *The deviation from this expectation for wild type A1-LCD is large both in the simulations (Fig. 2C) and the experiments performed in vitro (Fig. 3C) but very small for the mutant A1-LCD+12D. Why is the deviation from expectation for A1-LCD+12D so small, especially experimentally? The readers will greatly appreciate that the authors propose an explanation or, even better, that they address this question with a computational experiment.*

Response to comment 3: Please note that we already proposed an explanation for this phenomenon in the main text. Specifically, we stated that in the context of the FUS-LCD and A1-LCD +12D mixture, “Although the aromatic sticker interactions remain unchanged, the electrostatic interactions should be weakened. We reasoned that this would generate a dilute arm with a more convex shape, and this is precisely what we observe (Fig. 2E).” In conjunction with our schematic in Figure 2C, this suggests that switching from wild-type A1-LCD to A1-LCD +12D weakens the heterotypic interactions, causing them to be on par with the homotypic interactions, thereby resulting in the two-dimensional dilute arms observed in Figures 2E and 3D.

Comment 4: *The results obtained for A1-LCD+12D presented in Fig. 6 are those obtained experimentally. It would however be interesting to see the simulation results as well, as in Fig. 5; this will give the readers additional confidence that the simulations provide realistic representations of the condensates.*

Response to comment 4: We have now included these results in the SI. Importantly, our model predicts that A1-LCD +12D should have a slightly lower c_{sat} than FUS-LCD, while the opposite is true from experiments. Because of this, the slopes of the tie-lines for the simulations are different from those for the experiments. Thus, even though the exact values do not agree, the overall message underlying the meaning of the slopes of the tie lines remains the same across the simulations and experiments.

Comment 5: *The results presented in Fig. 8 are interesting and help understand those presented in Fig. 7C. The scenario represented in panel C of Fig. 8, where heterotypic interactions are weak, seems to correspond to the formation of two co-existing immiscible liquid phases. According to the authors this scenario should lead to a phase diagram where C_{sat} , C_{tot} and C_{dense} are not in a straight line. The authors should clarify this by computing the phase diagram of this system.*

Response to comment 5: The scenario represented in Fig. 8C corresponds to a system comprising three coexisting phases. Here, there is one dilute phase and two distinct dense phases, each with distinct protein concentrations. In this case, one would not expect a tie line, but rather a tie triangle (please see: <https://doi.org/10.1021/acs.chemrev.2c00814>) that connects the three phases together on the three-dimensional phase diagram. We recognize that our initial wording may have been misleading. We have now revised the original sentence to read “If phase separation gives rise to precisely two coexisting phases, then a single line should connect the three points.”

Comment 6: *It would be very helpful for the readers if the authors computed the full phase diagram for the A1-LCD FUS-LCD system from simulations. This would not generate new information but would make the work easier to follow for non-specialized readers of a journal such as Nature Communications.*

Response to comment 6: Please note that in Figure 2B, we show the full phase diagrams of various mixtures of FUS-LCD and A1-LCD as a function of temperature. Then, in Figure 5, we show the

initial, dilute, and dense phase concentrations of various mixtures of FUS-LCD and A1-LCD at a single temperature. So, what the reviewer requests was already furnished because we have low and high concentration arms of the coexistence curves.

Comment 7: *Contact maps of the simulations of 0, 50 and 100% FUS-LCD mixtures could be shown to strengthen the conclusion that heterotypic electrostatic interactions enhance phase separation.*

Response to comment 7: In our case, because we are treating electrostatic interactions using a mean-field model instead of a site-specific model, contact maps will not show clear signs that electrostatic interactions are driving phase separation. In lieu of contact maps, we have introduced a novel crosslinking analysis. We propose that this analysis is simpler to interpret and more generalizable to multiple components. As shown in Figure 7C, we find that FUS-LCD molecules are significantly more likely to interact with A1-LCD molecules than with other FUS-LCD molecules, suggesting an important role for heterotypic interactions.

Comment 8: *The ionic strength experiment in Fig. 3F is very nice and informative. However, the differences in the C_{dilute} of a 50% mixture are small. Maybe having a third condition (NaCl concentration higher than 300 mM) to see if the magnitude of the change in C_{dilute} increases further would help.*

Response to comment 8: We have now performed experiments with 600 mM NaCl and find a similar trend. Importantly, the change in C_{dilute} is not as important as the change in curvature of the curve connecting the C_{dilute} values. We find that as the salt concentration increases, the curvature decreases, suggesting the heterotypic interactions become more similar to the homotypic interactions as the electrostatic interactions are weakened, which is precisely what we expect. We have clarified this point in the main text.

Comment 9: *The manuscript is well-written but the figures need some work. Some of them could be moved to the Supplementary Material, some others could be merged and, in some cases, they could a little more clear: in any case 9 Figures appears excessive for Nature Communications.*

Response to comment 9: Please note that Nature Communications permits up to ten full size figures / display items. So, the figure count is not excessive and is in keeping with what the journal permits.

REVIEWERS' COMMENTS

Reviewer #1 (Remarks to the Author):

Thanks for the response. Overall the authors addressed most of my comments and made necessary edits to improve the manuscript. And I appreciate it. I have a few minor points to discuss with the authors.

Comment 2: Thanks for the clarification. I would appreciate it if you could share some references with mathematical proof on how the interplay between heterotypic and homotypic interactions determine the convex/concave shape the phase boundary. Also, based on what authors claimed in the rebuttal letter, the shape is determined by either homotypic or heterotypic interaction is dominant. It seems different than what the figure 2C said "positive or negative cooperativity between homotypic and heterotypic interactions". Personally I think the dominance is more correct and accurate. Think of an edge case where a phase separation system only has attractive heterotypic interactions. The phase boundary is of course concave. And that is not because of positive or negative cooperativity between heterotypic and homotypic. It is because there's only heterotypic interactions that's dominating the interactions.

Comment 3-6: Thanks for addressing my comments.

Comment 7: Thanks, now I see it. It is fascinating that the slopes of tie lines are determined by heterotypic and homotypic interaction interplays, as well as that the experiments align so well with simulations. I have a minor point which might help the manuscript, because not all readers are familiar with tie lines. If possible, could authors give some intuitive explanations for this phenomenon? For example, describe when A-A interaction dominates, what is the physical picture of the scenario, and what does slope < 1 means.

Reviewer #2 (Remarks to the Author):

The authors have addressed all my comments thoroughly. Their manuscript represents an important contribution to the field and I am happy to recommend it for publication.

Reviewer #3 (Remarks to the Author):

The authors have addressed most of my concerns, in some cases by carrying out additional work, and I think this very interesting manuscript can be published in Nat Commun without further changes.

Responses to comments of Reviewers

We are grateful to the three reviewers for their positive responses. Reviewers 2 and 3 were happy with all the changes we made. Reviewer 1 asked for two minor clarifications, and we have made the changes they requested. Our responses to their comments are noted below.

Comment 1: This was about comment 2 in our original response. *Thanks for the clarification. I would appreciate it if you could share some references with mathematical proof on how the interplay between heterotypic and homotypic interactions determine the convex/concave shape the phase boundary. Also, based on what authors claimed in the rebuttal letter, the shape is determined by either homotypic or heterotypic interaction is dominant. It seems different than what the figure 2C said “positive or negative cooperativity between homotypic and heterotypic interactions”. Personally I think the dominance is more correct and accurate. Think of an edge case where a phase separation system only has attractive heterotypic interactions. The phase boundary is of course concave. And that is not because of positive or negative cooperativity between heterotypic and homotypic. It is because there’s only heterotypic interactions that’s dominating the interactions.*

Response to comment 1: A computational treatment of the interplay between homotypic and heterotypic interactions was first introduced in our original LaSSI paper. A recent follow-up from the work of Deviri and Safran might be useful reading for the reviewer. In the revision, we cite this work and related studies via the following statements. “A concise treatment of some of the nuances underlying mathematical aspects of shapes of phase boundaries that result from the interplay between homotypic and heterotypic interactions and its effect on buffering has been described by Deviri and Safran⁷³. Their work provides a useful introduction to the concepts that drive our analysis, and it was based in part on the original work of Choi et al.,⁵¹ Riback et al.,⁴⁵ and Seim et al.,²⁸.” Also, we have deleted all the verbiage regarding positive versus negative cooperativity.

Comment 2: This was about Comment 7 in our original response. *Thanks, now I see it. It is fascinating that the slopes of tie lines are determined by heterotypic and homotypic interaction interplays, as well as that the experiments align so well with simulations. I have a minor point which might help the manuscript, because not all readers are familiar with tie lines. If possible, could authors give some intuitive explanations for this phenomenon? For example, describe when A-A interaction dominates, what is the physical picture of the scenario, and what does slope < 1 means.*

Response to comment 2: We have revised the main text to include the following sentences: “In the A-B mixture, a tie line with a slope that is less than unity will imply that the concentration of component A changes more significantly across the phase boundary than that of component B. This would suggest that component A, specifically homotypic interactions of component A, is the main driver of phase separation. Conversely, if tie lines have slopes that are greater than unity, then the homotypic interactions among B molecules are the stronger drivers of phase separation. Note that these pronouncements regarding numerical values of slopes of tie lines are governed by the choices of the axes we assign to concentrations of components A and B.”